# Deep Learning Test Platform for Maritime Applications: Development of the eM/S Salama Unmanned Surface Vessel and Its Remote Operations Center for Sensor Data Collection and Algorithm Development

Juha Kalliovaara [1,2,*], Tero Jokela [1], Mehdi Asadi [1], Amin Majd [1], Juhani Hallio [1], Jani Auranen [1], Mika Seppänen [1], Ari Putkonen [1], Juho Koskinen [1], Tommi Tuomola [1], Reza Mohammadi Moghaddam [3] and Jarkko Paavola [1]

[1] School of ICT, Turku University of Applied Sciences, 20520 Turku, Finland; tero.jokela@turkuamk.fi (T.J.); mehdi.asadi@turkuamk.fi (M.A.); amin.majd@turkuamk.fi (A.M.); juhani.hallio@turkuamk.fi (J.H.); jani.auranen@turkuamk.fi (J.A.); mika.seppanen@turkuamk.fi (M.S.); ari.putkonen@turkuamk.fi (A.P.); juho.koskinen@turkuamk.fi (J.K.); tommi.tuomola@turkuamk.fi (T.T.); jarkko.paavola@turkuamk.fi (J.P.)

[2] Department of Computing, University of Turku, 20014 Turku, Finland

[3] Independent Researcher, Mashhad 1696700, Iran; reza.mohammadi.me2@gmail.com

\* Correspondence: juha.kalliovaara@turkuamk.fi

**Abstract:** In response to the global megatrends of digitalization and transportation automation, Turku University of Applied Sciences has developed a test platform to advance autonomous maritime operations. This platform includes the unmanned surface vessel eM/S Salama and a remote operations center, both of which are detailed in this article. The article highlights the importance of collecting and annotating multi-modal sensor data from the vessel. These data are vital for developing deep learning algorithms that enhance situational awareness and guide autonomous navigation. By securing relevant data from maritime environments, we aim to enhance the autonomous features of unmanned surface vessels using deep learning techniques. The annotated sensor data will be made available for further research through open access. An image dataset, which includes synthetically generated weather conditions, is published alongside this article. While existing maritime datasets predominantly rely on RGB cameras, our work underscores the need for multi-modal data to advance autonomous capabilities in maritime applications.

**Keywords:** deep learning; multi-modal sensing; datasets; unmanned surface vessel; remote operations center; situational awareness; sensor fusion; open-access datasets; synthetic data; autonomous navigation

## 1. Introduction

Digitalization and increased autonomy in transportation are expected to advance significantly in the near future. This development has the potential to create more sustainable, safer, more efficient, and more reliable service chains, in line with the two global megatrends of digitalization and climate neutrality. With the constant growth of maritime traffic, safety and security are also of paramount importance. The autonomous operation of maritime vessels is globally recognized as a promising solution to address these safety and security concerns.

Unmanned surface vehicles (USVs) are boats or ships that operate on the water without a crew. They can be controlled remotely or autonomously and have various applications in civil and military fields, such as environmental monitoring, search and rescue, mine clearance, and anti-submarine warfare. USVs have several advantages over manned vessels, such as lower cost, smaller size, and higher efficiency. USVs also require remote observation

and a mechanism for remote control. As such, a Remote Operations Center (ROC) is a critical component of a USV.

This article describes the development of a test platform comprising a USV and an ROC at Turku University of Applied Sciences (Turku UAS). Our USV, named eM/S Salama, features an aluminum catamaran hull that measures 6.8 m in length and 3 m in width and operates using electric propulsion. Our USV is larger than the typically available research prototypes and can carry up to 10 persons. The catamaran hull provides a stable platform and can stand waves up to 2 m in height. Our ROC enables the control and monitoring of the USV remotely via wireless links, such as mobile networks or satellite connections, that offer sufficient data rate and latency. We present in detail the whole process of building the test platform.

The article highlights the challenges in enhancing autonomous features in maritime environments and the need for distinct deep learning applications tailored to specific settings. It also emphasizes the importance of securing adequate and relevant data for deep learning algorithms, which remains a primary challenge in the domain of autonomous maritime vessels. To address this, we introduce a multi-modal sensing and data collection system integrated into the USV eM/S Salama. Each sensor type has unique advantages and limitations, and no single sensor can guarantee sufficient reliability or accuracy in all different situations. Therefore, sensor fusion, which combines data from different sensors, is used to provide complementary information about the surrounding environment.

The aim is to create synchronized and annotated multi-modal datasets to be published in open access. The article presents our first published dataset. The first version of our sensing system includes RGB and thermal cameras, stereo vision camera, and a light detection and ranging (LiDar) sensor. The data collected are stored in a data platform in a format that is readily usable for the development of deep learning algorithms. These data are crucial for the advancement of deep learning and computer vision techniques, which are employed to gain situational awareness of the USV's environment. The acquired situational awareness plays a key role in providing decision support and facilitating autonomous navigation decisions.

This paper is organized as follows. Section 2 reviews the existing research on USVs, exploring established research prototypes and pertinent regulations governing USVs. Section 3 gives a detailed description of our test platform and the construction of the USV and its associated ROC. Section 4 provides an in-depth look at our data collection system, data storage platform, and the data that we have collected. It also outlines the methodologies we have employed for data annotation and showcases a selection of the deep learning algorithms we have developed. Section 5 discusses the significance of our test platform and outlines our ambitious plans for its future expansion. The paper concludes with Section 6, which summarizes the key points and findings.

## 2. Background

Maritime transportation is currently undergoing a shift towards digitalization and automation [1]. Digitalized and automated operations and autonomous navigation can help in mitigating accidents due to human errors [2]. However, the role of humans is constantly increasing in the design phase of autonomous shipping systems [3].

International Maritime Organization (IMO) has defined four levels of autonomy for USVs [4]. The first level provides automated processes and decision support, while the second is a remotely controlled ship with seafarers onboard and the third is a remotely controlled ship without seafarers onboard. The fourth level is a fully autonomous surface vehicle. To enable full autonomy, for example, methods to define the ship state, smart path planning and navigation, collision avoidance, remote monitoring and control, wireless communication systems, sensors, and deep learning algorithms to provide situational awareness are needed. The development of autonomous shipping requires a delicate balance between technological advancements and safety considerations, ensuring the well-being of both seafarers and the environment [5].

### 2.1. USV Research and Projects

Many researchers and organizations around the world are working on improving the performance and functionality of USVs, and many prototypes have been successfully tested and deployed. The amount of research and trials conducted on USVs has increased significantly, especially during the past 10 years, as we are now starting to have the technologies to build USVs available at a reasonable cost [6–12].

However, USVs are not yet at a phase where they are in wide commercial use. The automation of processes in shipping is constantly increasing, but remotely controlled or autonomous vessels are not yet seen in normal traffic [13]. More trials are needed to see if USVs are ready for industrial practice and if they can be used in a safe and efficient manner. The trial results will also contribute to the development of the relevant regulation. A comprehensive comparison on the past and recent USV surveys, projects and prototypes is available at [14].

In Europe, the Partnership on Zero-Emission Waterborne Transport under Horizon Europe aims at the large-scale introduction of resilient and secure autonomous operations by 2050. In their vision, digitalization will lead to a higher degree of automation and autonomy, automated and autonomous systems, ship operations (both maritime and inland navigation), and remote control from the shore by 2030 [15].

The Advanced Autonomous Waterborne Applications Initiative (AAWA) [9] was a project launched by Rolls-Royce to explore the economic, social, legal, regulatory, and technological factors that need to be addressed to make autonomous ships a reality. The project aimed to answer critical questions such as what technology is needed and how it can best be combined to allow a vessel to operate autonomously, miles from shore. To make the case for autonomous ships, it was important to consider how an autonomous vessel can be made at least as safe as existing ships, what new risks it will face, and how they can be mitigated.

The AAWA architecture is widely used in current USVs, and thus, we have also chosen it as a basis for designing our own USV functional architecture, which is represented in Figure 1.

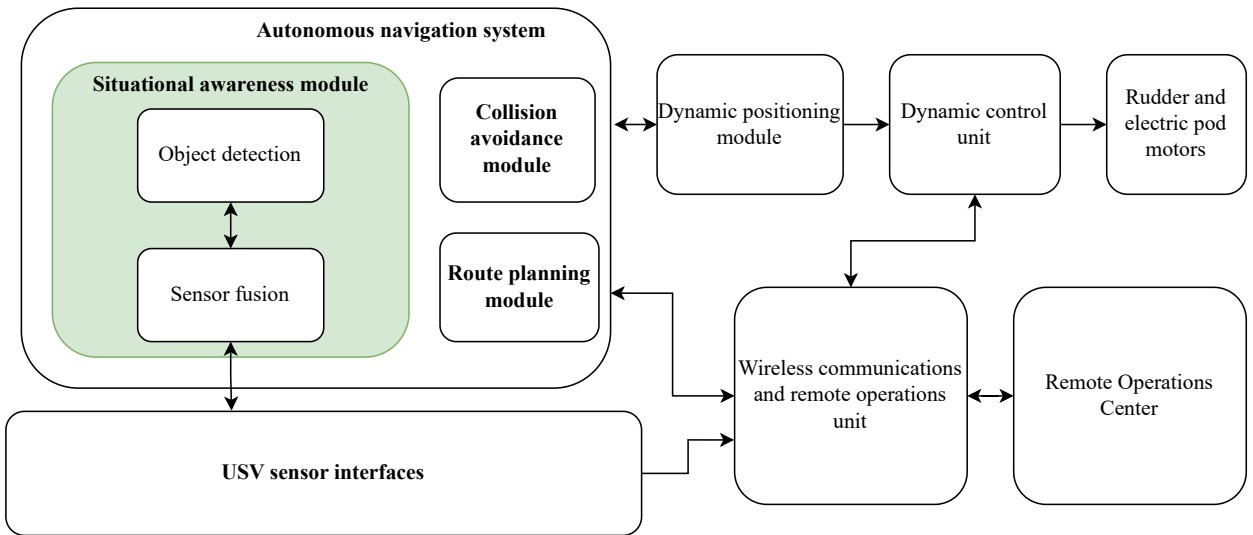

**Figure 1.** The USV functional architecture.

The figure illustrates the main components that are required for autonomous and remote operations. The autonomous navigation system exploits the vessel sensing system for the observation of the vessel surroundings and has the intelligence that the vessel needs to have to operate autonomously.

The detection of the obstacles is performed on fused sensor data using deep learning algorithms. The route of the vessel is defined in the route planning module where the

waypoints are defined, and the situational awareness module, together with the collision avoidance module, applies changes to the route based on observations from the sensor system. The dynamic positioning module automatically maintains the vessel's position and heading by using its own motors and rudders through a dynamic control unit (DCU) we have developed.

A reliable wireless link between the vessel and the ROC is required to enable remote monitoring and control. As vessels can move in diverse environments, hybrid connectivity systems are considered the best solution. Hybrid connectivity translates into utilizing several different connectivity networks (e.g., 4G, 5G, VHF Data Exchange System (VDES), Satellite) for data transmission. The wireless communications module also contains the intelligence needed to pass the control commands from the ROC to the DCU.

In light of the USV research developments [16], it is clear that the future of shipping is on the verge of a major transformation. The advancements in USV technology and the increasing automation of shipping processes are paving the way for a new era of maritime operations. However, the journey toward the wide commercial use of USVs is still ongoing. It requires not only technological innovation but also a thorough understanding and revision of the existing regulatory framework [17].

The trials conducted on USVs and the research dedicated to improving their performance and functionality are crucial steps toward this goal. They provide valuable insights into the practical aspects of USV operation and highlight the areas where further improvements are needed. The work performed by various researchers and organizations worldwide, as well as initiatives like the AAWA, is essential in this field.

*2.2. Regulation*

Regulation is crucial to ensure the safety, efficiency, and standardization of the USVs. The IMO is the United Nations specialized agency responsible for the safety and security of shipping and the prevention of marine and atmospheric pollution by ships. The IMO is the main body that needs to update its regulatory framework to facilitate the integration of new technologies required for USVs. In their terminology, USVs are referred to as Maritime Autonomous Surface Ships (MASS).

The integration of new technologies like USVs into existing regulatory frameworks necessitates Regulatory Scoping Exercises. These exercises ensure that as technology evolves, so do the regulations governing their use, ensuring safety, security, and environmental protection. The IMO has conducted a regulatory scoping exercise on MASS that aimed to assess existing IMO instruments to see how they might apply to ships that utilize varying degrees of automation. The scoping exercise concluded that the MASS can be accommodated in the regulation with some modifications, the most critical issues being the role and responsibility of the master and the remote operator, questions of liability, consistent definitions of MASS, and the carriage of certificates [4].

Several committees within the IMO, including the Maritime Safety Committee, the Facilitation Committee, the Legal Committee, and the Marine Environment Protection Committee, are working on different aspects of USVs. The IMO plans to adopt a non-mandatory goal-based MASS Code by 2025, which will form the foundation for a mandatory goal-based MASS Code, expected to be enforced starting 1 January 2028.

The Convention on the International Regulations for Preventing Collisions at Sea (COLREGs) [18] is a set of rules published by IMO that must be adhered to by the ships to avoid collisions at sea. The USVs must thus also follow the COLREGS to ensure safety. The International Convention for the Safety of Life at Sea (SOLAS) [19] is an international maritime treaty established by the IMO. SOLAS sets out minimum safety standards in the construction, equipment, and operation of merchant ships.

The International Association of Marine Aids to Navigation and Lighthouse Authorities (IALA) is considering the introduction of MASS on the seas, as the autonomous vessels may benefit from different kinds of aids to navigation than the traditional ships [20]. For instance, digital announcements may be more suitable than voice-read navigational an-

nouncements. The International Hydrographic Organization (IHO) ensures that all the world's navigable waters are surveyed and charted. They are instrumental in standardizing the digitalization of charts and navigational information required for USV operations.

The United Nations Convention on the Law of the Sea (UNCLOS) provides the legal framework governing all activities in oceans and seas. For USVs, adherence to UNCLOS ensures that the vessels operate within internationally agreed-upon maritime laws, respecting territorial waters and exclusive economic zones [21].

In conclusion, the development and operation of USVs are governed by a complex web of international maritime laws and regulations. Adherence to these laws ensures the safe, secure, and environmentally friendly operation of USVs, paving the way for their increased use in research and other applications.

The establishment of USV test platforms is crucial in advancing these regulations. These platforms provide a practical environment for testing USVs' compliance with existing laws and identifying areas where the laws may need to evolve to accommodate the unique operational characteristics of USVs. ROCs, meanwhile, offer a controlled environment for monitoring and managing USVs, thereby ensuring their safe and effective operation. USV test platforms and ROCs are thus essential not just for the advancement of USVs but also for the evolution of the regulatory framework that governs them.

## 3. Test Platform

The development of USVs has been driven by the advancement of science and technology, as well as the increasing demand for marine vessels in different domains. USVs face many technical challenges, such as autonomous navigation, wireless communication, and collision avoidance. In this context, Turku University of Applied Sciences has designed and built a test platform, consisting of a USV and ROC, to support the research community and industry in efficiently utilizing deep learning to advance autonomous maritime vessel development.

This real-world platform enables development and performance verification of deep learning algorithms, helps bridge the digitalization skills gap, and fosters collaboration between academia and industry. The initiative is anchored in the Applied Research Platform for Autonomous Systems (ARPA), a testbed for maritime automation, autonomy, and remote control, thereby supporting the blue growth sector in Southwest Finland [22].

In order to ensure the platform's effectiveness, we have developed and built the test platform ourselves, using and integrating commercial off-the-shelf equipment into the overall system when available. A significant effort has been put into the development of the DCU, as described in Section 3.2. The development process has allowed us to gain a comprehensive understanding of every aspect of the operation of the USV. The goal has been to establish a research platform rather than develop a product for commercial purposes. The USV itself can currently be operated manually and remotely, while autonomous features are under development, with trials planned for 2024. Compared to other USVs built for research purposes, ours is the largest, to the best of our knowledge [14].

With this test platform in place, it is possible to research the challenges and opportunities of applying deep learning to remote sensing problems in complex marine environments. Section 4 introduces our efforts in this area.

### 3.1. Unmanned Surface Vessel

This section outlines the development of our USV eM/S Salama. The USV is a commercial craft with electric propulsion, the first in Finland.

In the following list, we go through the different phases of the vessel's development. The following sections give a more detailed description of the most relevant topics.

1. Defining requirements and specifications. The physical characteristics of the vessel are, for example, size, weight, speed, and powertrain. We chose an aluminum catamaran hull, which is 6.8 m long and 3 m wide and has a cabin. The chosen hull is Alpo Pro Boats MAX68 [23]. The boat has electric propulsion with two 10 kW electric E-tech POD 10 motors [24] installed to the stern of the boat. There are also rules and regulations that we must comply with to operate safely and legally. This also involves the vessel certification process, which is explained in Section 3.1.4.

2. Design the electrical system. We have several different electronic devices requiring 12/24/48 Volts Direct Current (VDC) and 230 Volts Alternating Current (VAC) electricity. There are several non-trivial safety, interference, and grounding issues that had to be solved to make the system work in a safe and efficient manner. We also designed and built a charging system and procured a generator as a backup energy option. The electricity systems are described in Section 3.1.1.

3. Designing the The Controller Area Network (CAN) bus architecture: CAN is a critical component, which is essential for communication between the vessel devices. Our devices use NMEA2000, J1939, and CANopen messages. We designed and built the overall CAN architecture from scratch. The CAN bus architecture is described in Section 3.1.3.

4. Simulations. Extensive simulations were performed for the overall vessel NMEA2000 system and the vessel's physical properties, including friction resistance, wave-making resistance, and air/wind resistance.

5. Procuring the hull, motors, and batteries. These were purchased from commercial suppliers. Special attention was given to ensuring compatibility between the components and the overall USV system design, including the remote and autonomous operation modes. The acquired batteries are used to power the propulsion system, as described in Section 3.1.1. The total capacity of the eight acquired lithium iron phosphate batteries is 34 kWh.

6. Designing and implementing a DCU. The DCU allows us to control the motors and the rudder with NMEA2000 messages to allow remote and autonomous operations. Normally, the motors only take analog voltages as their input to change their speeds. The DCU design is described in Section 3.2.

7. Selecting and integrating the sensors for situational awareness. This includes identifying the appropriate sensors to capture relevant data for situational awareness for both remote and autonomous operation of the vessel. The devices are described in Section 3.1.2.

8. Designing and implementing the ICT subsystem. The ICT subsystem facilitates data processing, storage, and communication within the vessel devices. The vessel Ethernet architecture is described in Section 3.1.3.

9. Implementing the electrical system. During this phase, we installed and integrated the electrical system components, including batteries, power distribution panels, wiring, and safety devices. We followed electrical and safety standards during the installation process. Thorough inspection measurements were conducted by a certified electrician.

10. Implementing the CAN bus and connecting all the relevant electronic systems, including propulsion, navigation, sensors, and control systems.

11. Performing extensive system testing and ensuring that the vessel meets all relevant safety regulations, certifications, and guidelines. This includes also preparing the documentation for the vessel certification and registration. To our knowledge, this is the first commercial craft in Finland using electric propulsion.

The USV is called eM/S Salama and is illustrated in Figure 2. The name "Salama" has roots in Arabic and African languages, and it translates to safety and security, which aligns well with the main aims of USVs. In Finnish it translates to lightning, which also aligns well with our eco-friendly electric vessel. In the following sections, we describe the USV in more detail.

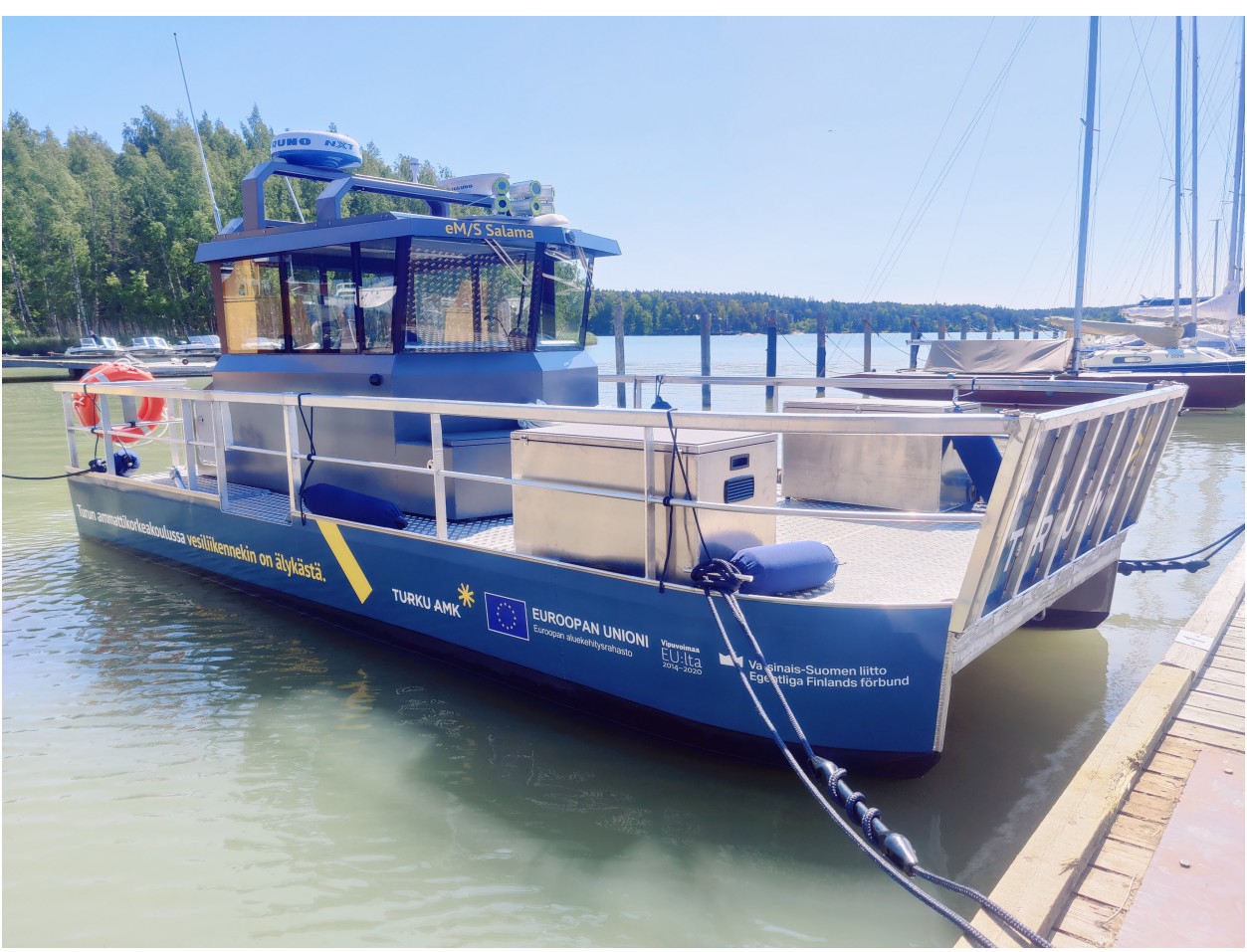

**Figure 2.** Unmanned Surface Vessel eM/S Salama.

### 3.1.1. Electricity

The USV is equipped with a sophisticated electrical system, as depicted in Figure 3. The propulsion system of the vessel relies on 48 VDC engines, supported by a 48 VDC battery system with a substantial storage capacity of 680 ampere-hours (Ah). To charge the main batteries, a 230 VAC system is employed. Power is supplied to the vessel through a specially designed inlet, which features an LED indicator for live voltage. This power can be sourced from a standard power grid at the harbor or from the vessel's onboard 3 kilovolt-ampere (kVA) generator. The existing charging system operates at a power of 3 kW, necessitating over 10 h to fully charge the battery from a completely depleted state. Upgrades to the charging system will be implemented as required. With the current battery setup, the electric propulsion system can maintain a speed of approximately 5 knots for a range of roughly 35 nautical miles, or the equivalent of 65 km. In our current system, we maintain a power reserve of 3 kW for the ICT subsystem and multimodal sensing. To date, we have employed generic Intel architecture professional laptop machines for data collection from the sensors. The overall power consumption during data collection has remained below 1 kW. However, as we anticipate that future autonomous navigation features will significantly increase power requirements, we are proactively planning to expand our battery setup when the need arises. Additionally, when we commence the autonomous navigation trials, we intend to augment the ICT subsystem with a 19-inch rack containing substantial computing resources.

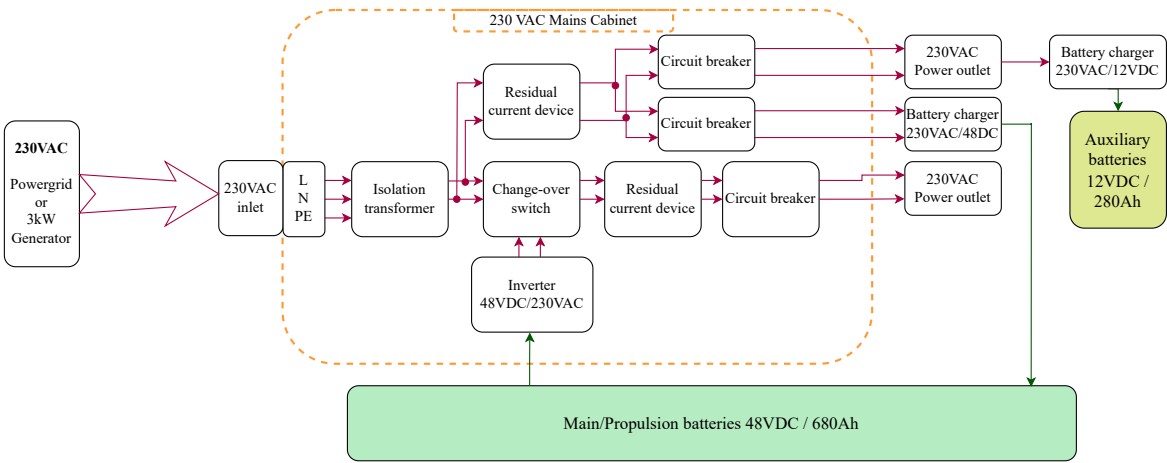

**Figure 3.** The USV electricity system.

To ensure safety and efficiency, all incoming power is isolated using a transformer. The 230 VAC power is utilized within the vessel to charge both the main and auxiliary batteries. In addition to this, power outlets have been installed inside the cabin to accommodate other equipment. These outlets can be powered by a 48 VDC to 230 VAC inverter when the external 230 VAC main is not connected. A change-over switch has been incorporated into the system to prevent the inverter power from being used to charge the batteries.

The propulsion system of the vessel is powered by eight 12 VDC lithium iron phosphate (LiFePO4) batteries. These batteries are arranged in two serial sets of four, forming 48VDC circuits that are connected in parallel. These sets of batteries are strategically placed on each side of the catamaran-type vessel housed within the pontoons. The batteries are further divided into four compartments to optimize weight distribution and ensure vessel stability. The compartment locations can be seen in Figure 4.

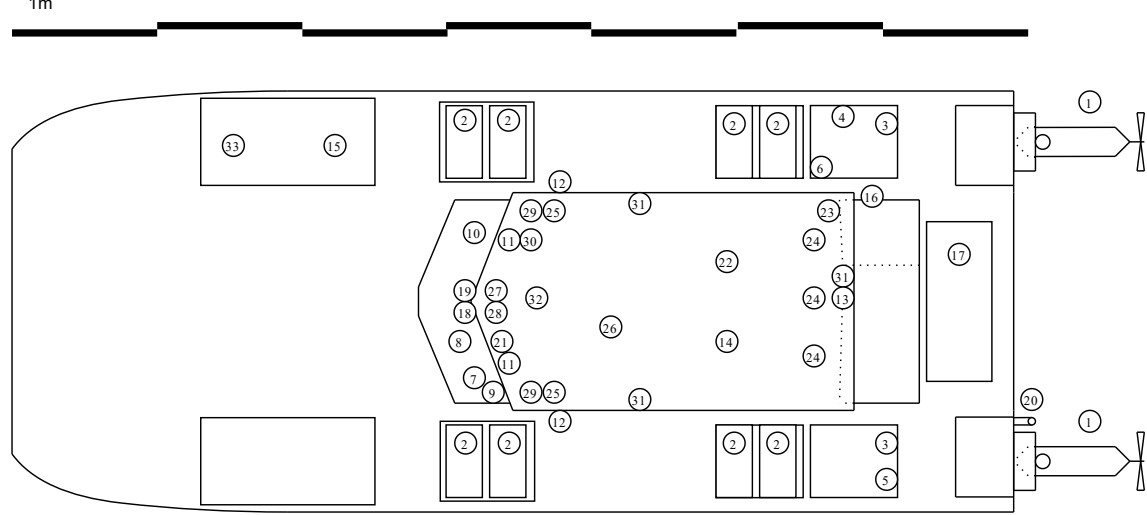

**Figure 4.** Equipment placement on the roof and on the deck of eM/S Salama.

The 48 VDC power from the main batteries also serves to supply the vessel's 12 VDC auxiliary system. The 12 VDC auxiliary batteries power the navigational equipment and other essential functionalities such as lighting, VHF radio, and heating systems. To accommodate the use of external 230 VAC equipment onboard, an inverter has been installed. This inverter utilizes the 48 VDC power when the vessel is offshore and the

mainland power grid is inaccessible. This comprehensive electrical system ensures the vessel's efficient operation while maintaining safety standards.

### 3.1.2. USV Devices

The USV incorporates several devices that are necessary for different purposes. This section describes the devices in the USV. The devices can roughly be divided into four groups, that is, the equipment for power and propulsion, navigation and communication, sensory and imaging, and safety, security and utility. The devices are listed in Tables 1 and 2, and their locations on the USV are further illustrated in Figures 4 and 5.

The power and propulsion equipment includes the electric motors, batteries, and related control systems. The navigation and communication equipment incorporates the devices that are on one hand required by the regulation (e.g., VHF radio, navigational lights, etc.) and on the other hand used for operating the vessel (autopilot, Global Positioning System (GPS), wireless data connectivity, chart display, etc.). Sensory and imaging devices are not strictly necessary for the manual operation of the vessel but are required for the remote and autonomous operations (cameras, LiDAR, radar, etc.). Further, this equipment is used for the data collection of datasets for deep learning purposes. Safety, security, and utility devices include equipment such as cabin heaters and anchors.

**Table 1.** Equipment listing on the roof and on the deck of eM/S Salama.

| Number | Name |
|--------|------|
| 1 | Electric motors |
| 2 | Propulsion batteries ($\times$8) |
| 3 | Motor inverters |
| 4 | Main relay and battery interface box |
| 5 | Propulsion battery charger (230 VAC to 48 VDC) |
| 6 | Main 48 V power switch |
| 7 | 12 V operating battery |
| 8 | 12 V emergency battery |
| 9 | 12 V operating battery current shunt |
| 10 | Diesel cabin heater |
| 11 | Windshield wipers |
| 12 | Navigation lights |
| 13 | Masthead light |
| 14 | Airmar 150WX Weather station [25] |
| 15 | Auxiliary Generator |
| 16 | Shore power plug (230 V) |
| 17 | Autopilot hydraulic pump |
| 18 | Horn |
| 19 | Deck light |
| 20 | Sonar transducer and speed meter |
| 21 | GPS antenna for VHF |
| 22 | Radar radome |
| 23 | VHF and AIS antenna |
| 24 | LTE and 5G communications antennas ($\times$3) |
| 25 | GPS antennas for GNSS+IMU |
| 26 | SC33 GPS compass for navigation [26] |
| 27 | RGB camera array ($\times$3) |
| 28 | Thermal camera array ($\times$3) |
| 29 | Cameras for stereo ($\times$2) |
| 30 | Camera array for ROC |
| 31 | Widefield cameras for ROC ($\times$3) |
| 32 | LiDAR |
| 33 | Anchor |

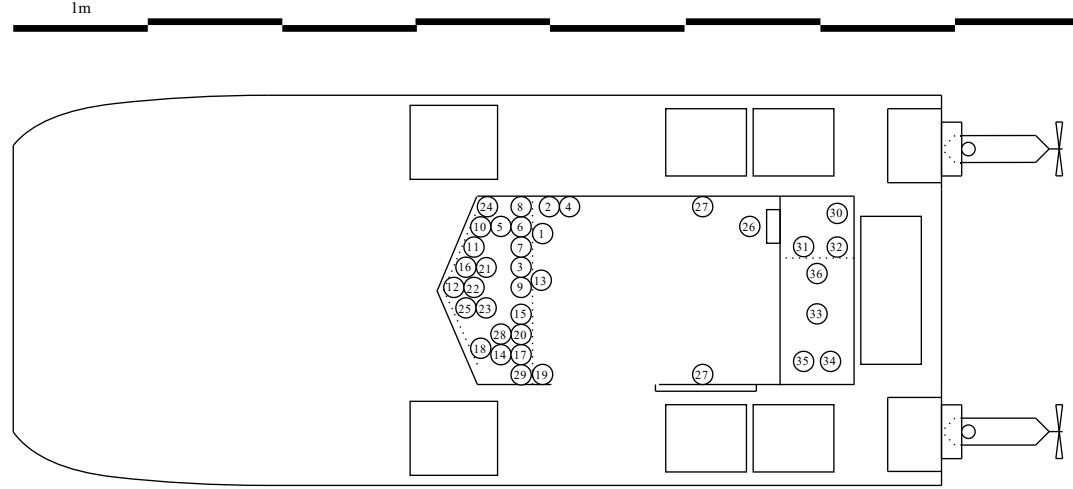

**Figure 5.** Equipment placement in the cabin of eM/S Salama.

**Table 2.** Equipment placement for the cabin of eM/S Salama.

| Number | Name |
| --- | --- |
| 1 | Boat's wheel |
| 2 | Motor throttles |
| 3 | Boat's electrical switches |
| 4 | Motor power switches |
| 5 | Motor status displays |
| 6 | Tilt and trim switches |
| 7 | EPEC multifunctional display [27] |
| 8 | Furuno autopilot |
| 9 | Furuno multi display |
| 10 | Furuno sonar |
| 11 | Furuno interface box |
| 12 | Ethernet Switch for Furuno Navnet |
| 13 | Navigation 17" touchscreen display |
| 14 | Ethernet Switch for navigation/ROC PC |
| 15 | Marine VHF Radio |
| 16 | class-B AIS transceiver |
| 17 | FM Radio |
| 18 | Electrical switchboard for low voltage |
| 19 | 12 V Main switches for cabin |
| 20 | 12 V battery monitor |
| 21 | Battery charger AC-to-12 V |
| 22 | DC/DC 48 V-to-24 V |
| 23 | DC/DC 48 V-to-12 V |
| 24 | VHF antenna isolator |
| 25 | CANBUS switch/bridge |
| 26 | Electrical switchboard for mains AC |
| 27 | AC power outlet |
| 28 | USB charger outlet |
| 29 | Heater controller |
| 30 | 48 V switches for cabin and sensoring equipment |
| 31 | AC Isolation transformer |
| 32 | Inverter 12 DCV-to-230 VAC |
| 33 | Computer and networking rack |
| 34 | Server space power switchboard |
| 35 | Server space power supplies |
| 36 | Smartbox for 4G/5G connectivity [28] |

### 3.1.3. Networking

The USV is equipped with a sophisticated network system based on Ethernet and CAN based NMEA2000 bus, which are described in this section. The Ethernet networks are used for the data exchange for the devices requiring high data rates, such as camera sensors, while NMEA2000 is used for the interconnection of NMEA2000-capable devices with lower data rate requirements. The Ethernet-based networks are primarily divided into three separate, function-specific networks:

1.  The Vessel Navigational Network: This network is designed for manual use and comprises a switch connected to devices such as radar and sonar. It also includes a computer running the navigation software. A touchscreen display is utilized for controlling and viewing the navigation system. Additionally, the navigation system is integrated with the vessel's NMEA2000 bus, ensuring seamless communication and data transfer.

2.  The Remote Operation Network: This network is equipped with cameras connected via a network switch that provides Power over Ethernet (PoE). These cameras offer a comprehensive 360-degree view. The display of the navigation computer is duplicated and transferred as a Real Time Streaming Protocol (RTSP) stream for remote operators' observation. The network also includes a firewall and security gateway to create a secure transmission tunnel to the ROC. Wireless connection modems create an aggregated link over three commercial mobile network operators and, in the future, also over satellite connectivity. The ROC is described in detail in Section 3.2.

3.  The Sensor Network for Autonomy: This network comprises several different types of cameras, LiDAR, and an OXTS Inertial Measurement Unit (IMU) [29] for deep learning to create situational awareness for autonomous operation. The sensor network can be connected to the ROC for control and security purposes. The sensor network includes a LiDAR with a GPS-assisted IMU that produces a point cloud of the near surroundings. Stereo cameras are used to detect objects and their distances. Three RGB video cameras and three thermal cameras are stitched together to provide a 180-degree front view.

These networks collectively ensure the efficient operation and control of the USV, whether it is under manual use, remote operation, or autonomous operation. The USV Ethernet architecture is illustrated in Figure 6.

NMEA2000 is a communications standard used for connecting marine sensors and display units within ships and boats. It allows any sensor to talk to any display unit or other device compatible with NMEA2000 protocols. The communication is based on CAN bus technology, widely used on vehicles and fuel engines.

Various instruments that meet the NMEA2000 standard are connected to one central cable, i.e., a backbone. The backbone relays data among all of the instruments on the network. This allows one display unit to show many different types of information and allows the instruments to work together since they share data. NMEA2000 allows devices made by different manufacturers to communicate with each other.

The vessel's main NMEA2000 bus is illustrated in Figure 7. It connects these devices for basic operation: Furuno Display, Furuno Autopilot, Automatic Identification System (AIS) transceiver, VHF Radio, SC33 GPS Compass, Airmar 150WX Weather Station, two Yacht Devices YDCC-04 Circuit controls[30], a computer that operates navigation software, and EPEC multifunctional display. For autonomous and remote operation, the bus also has a DCU that is NMEA2000-compatible. The DCU is designed and built in-house and described in more detail in Section 3.2. The DCU has three devices: DCU commander, which is a gateway/controller for ROC; DCU Motor, which controls the motors by replacing throttle levers when in use; and the DCU Rudder, which controls the autopilot remotely. Shipmodul Miniplex [31] is forwarding NMEA2000 messages to the ROC. NMEA2000 to Robotic Operating System 2 (ROS2) bridge is used for adding NMEA2000 capabilities to the ROS2 network, which is used for autonomous operations.

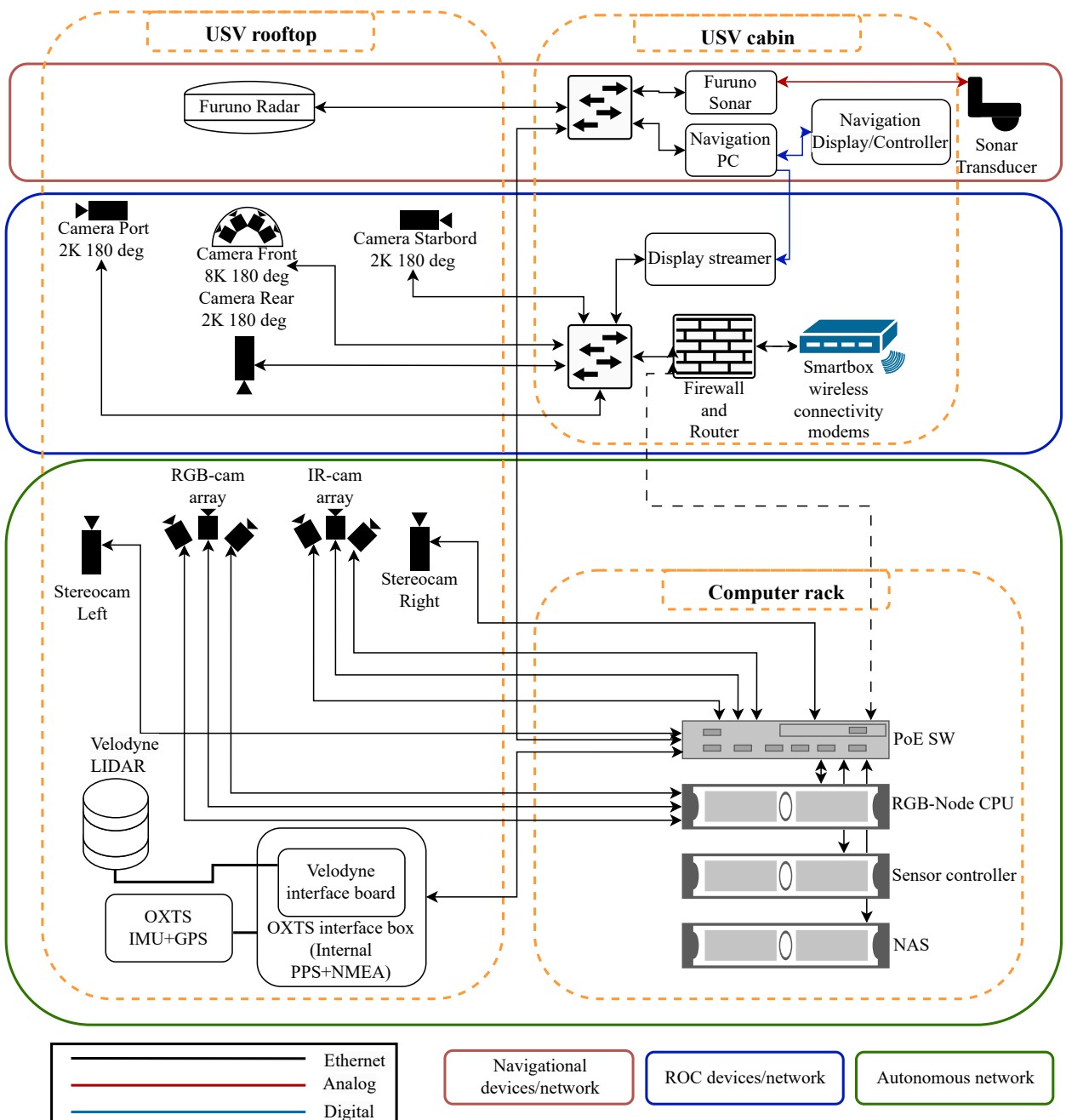

**Figure 6.** The USV Ethernet architecture.

The vessel also has other CAN-based buses besides NMEA2000. CANopen is used for the management of main batteries. One bus is used for controlling/using the battery interface box (BIB), and the other is used for connecting individual batteries to the BIB. There are also two point-to-point J1939 busses, which send motor information to the E-Tech displays from the E-Tech motor inverters.

The BIB is connected to an EPEC multifunctional display, which functions as a battery monitor in accordance with regulations. A user interface was developed using the Codesys development environment and the OpenBridge Design Guideline [32]. The latter is a free resource for user interface and workplace design, specifically adapted to the maritime context and its regulations.

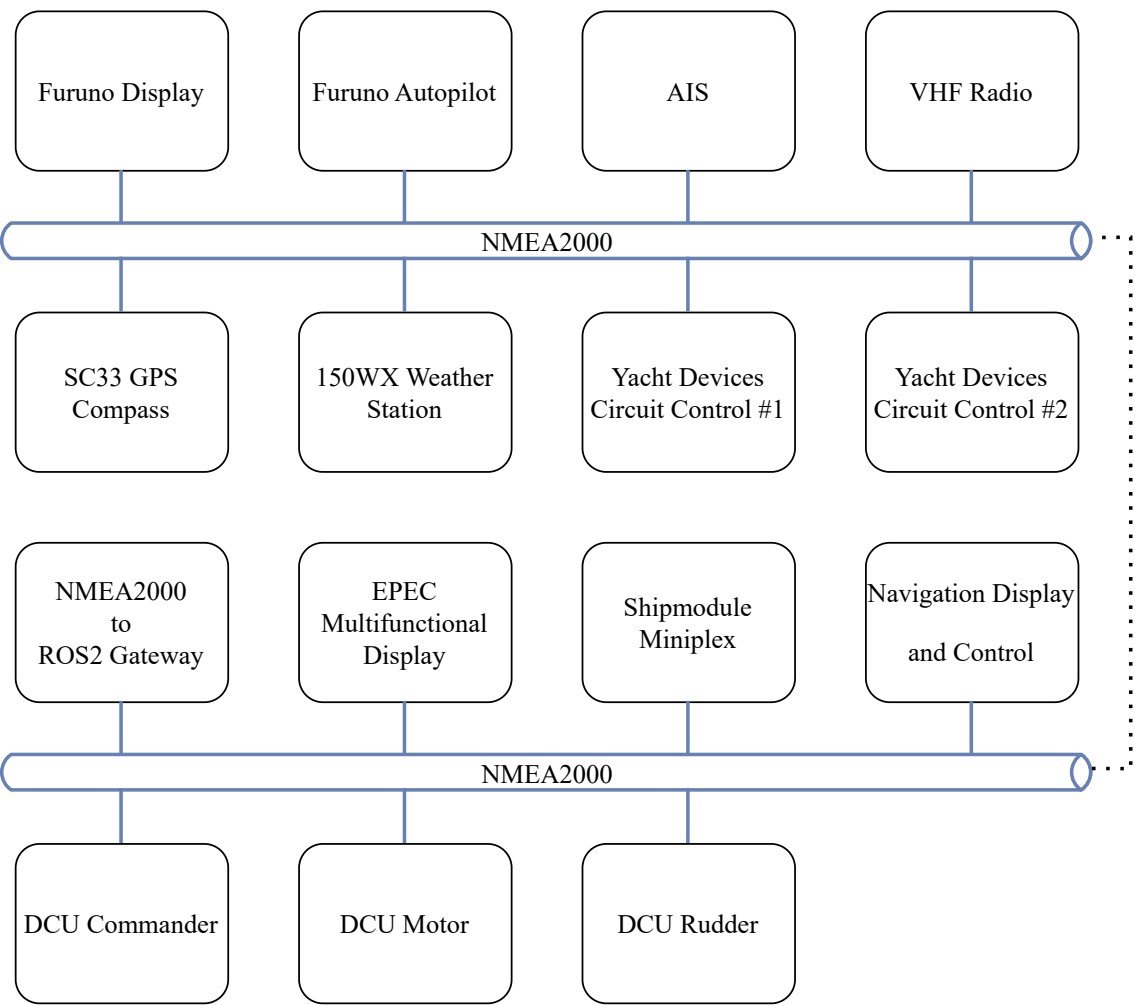

**Figure 7.** The USV NMEA2000 bus architecture.

### 3.1.4. Commercial Craft Certification in Finland

To use the USV for research, it must first be certified and registered as a commercial craft. This is performed by Eurofins in Finland. Each craft requires individual certification. The Finnish commercial craft certification guidelines ensure safety and environmental sustainability for work boats 5.5 to 24 m long [33].

The certification process involves checking compliance with rules using technical drawings before construction, inspecting the ship hull during construction, and conducting a test drive and stability test to determine the draft and center of gravity. An inspection report is submitted to the Finnish Transport and Communications Agency (Traficom) for commercial craft approval.

After certification, the vessel is registered as a commercial craft and inspected, and a license for the VHF radio is obtained along with the Maritime Mobile Service Identity (MMSI) for the radio station. The USV is then ready for research use.

Our USV is certified as a category C watercraft, which means it is designed for use in coastal waters, large bays, estuaries, lakes and rivers where wind conditions up to and including Beaufort force 6 and significant wave heights up to and including 2 m may be experienced [34].

### 3.2. Remote Operations Center

A ROC for a USV, when integrated with deep learning algorithms, is a potent tool that brings numerous advantages. Deep learning algorithms can analyze and process precise

situational data collected from the USV multi-modal sensoring system and increase the efficiency and safety of the USV. Efficiency improves by optimizing processes, and safety improves when people can be removed from dangerous operating environments or when human errors can be reduced in operations.

In the ROC, operators can monitor, support, assist, supervise, and control the USV. The ROC can monitor the USV or directly control the USV's systems. The key components for the remote operation of the USV are the USV multi-modal censoring system, data communication links, and the ROC itself. The USV multi-modal sensoring system provides information about both the USV itself and its environment. The sensor data of the USV transferred to the ROC can also be utilized directly in a digital twin of the vessel.

The ROC is shown in Figure 8. The display setup consists of five 55-inch screens and three 24-inch screens for 180-degree front and back views, telemetry data, and map data. Additional screens can be added, for example, to show information from other sensors, such as side-view, thermal cameras, and LiDARs. Figure 8 shows that have two additional screens to show the view from both sides of the vessel. Tablets are used to control the vessel, as this allows an easier way to modify and test different control systems than having a physical controller or an exact replica of the vessel dashboard. The telemetry data from the vessel are transmitted with NMEA2000 messages and the video streams using RTSP.

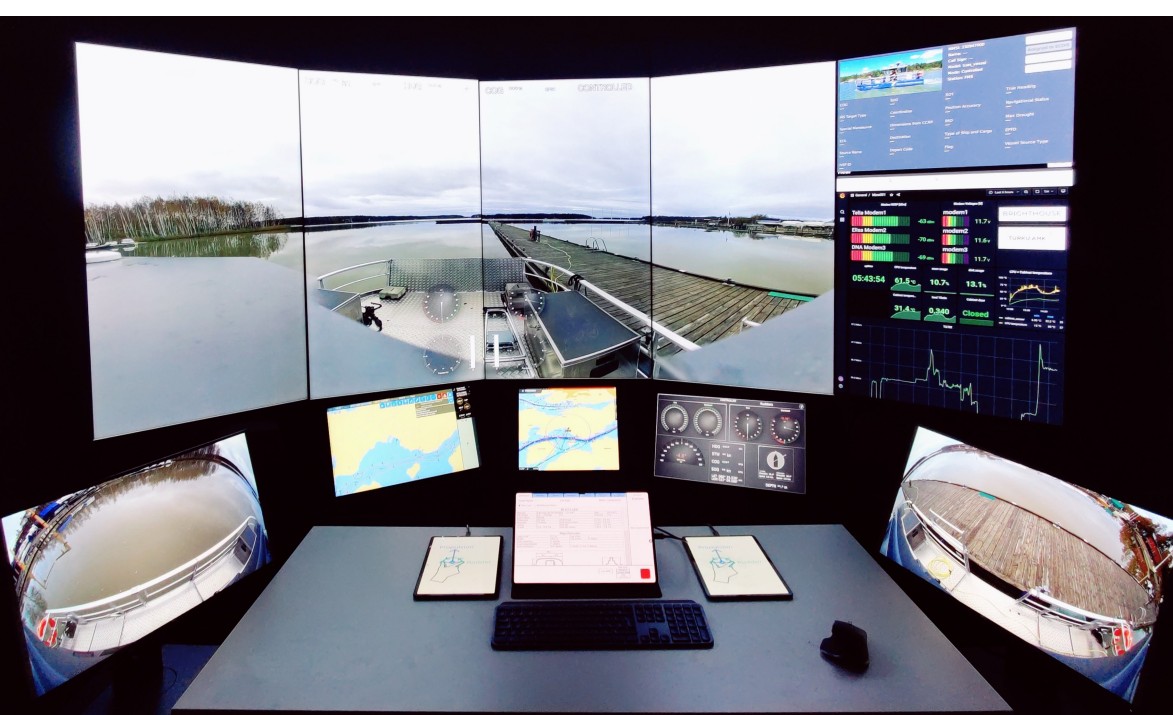

**Figure 8.** Turku UAS Remote Operations Center.

The ROC software was procured from a commercial supplier. Our main focus and contribution in the ROC development has been on the DCU, whose development is described in more detail. The ROC itself is described in the article on a detail level, which gives an understanding of how it functions and how the DCU is integrated into the overall ROC system. Figure 9 shows a high-level schematic of the ROC implementation. The data from vessel device interfaces, such as cameras, AIS, radar, LiDAR, and control systems, are transmitted to a data processing and communication unit. The data are parsed and transformed into a suitable format so that it can be visualized and used by the ROC software. The communication unit transmits the data over a wireless link, such as a mobile network or satellite connection, to the ROC. The wireless link telemetry and control data are transmitted through Google Cloud services, while the vessel telemetry, control data, and video streams are transmitted directly through a secure IP Security Architecture (IPsec)

tunnel. Both the vessel and the ROC have a firewall. The ROC central computer processes the received data and shows it on the ROC display setup.

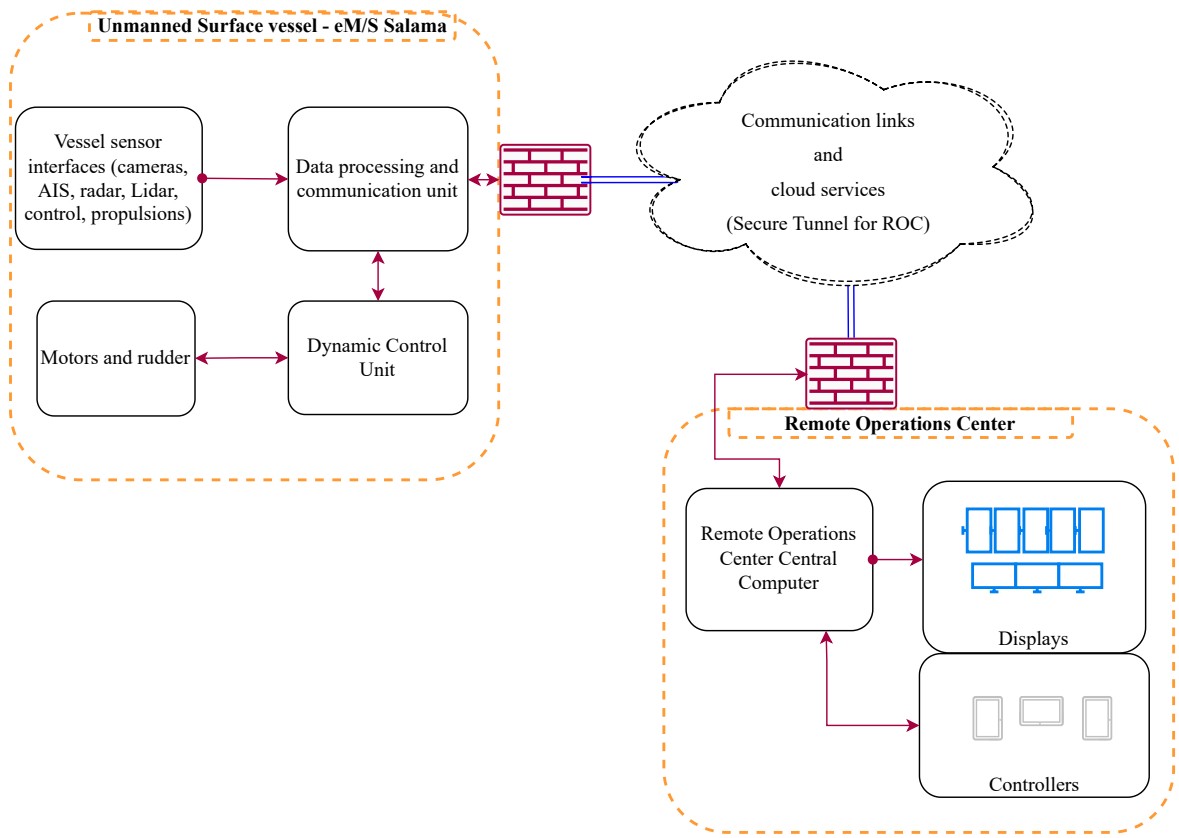

**Figure 9.** Remote Operations Center operational architecture.

The USV DCU system, as depicted in Figure 10, is composed of several components: the ROC, the DCU Commander, the DCU Motor, the DCU Rudder, and other NMEA2000 devices.

The ROC communicates with the DCU Commander via NMEA0183 messages, sending commands to change the rotations per minute of the motors and the rudder angle. The DCU Commander then communicates these NMEA2000 control messages to the DCU Motor and DCU Rudder.

The DCU Motor and DCU Rudder are two devices with custom-printed circuit boards. The DCU Motor controls the motors by mimicking throttle inputs, while the DCU Rudder controls the rudder as a remote for the Furuno autopilot. Both devices incorporate a CAN controller for NMEA2000, a digital-to-analog converter (DAC), and relays. The DACs provide throttle input and the rudder remote input, while the relays are used for changing the throttle input provider and as a switch for the autopilot.

Status messages are shared between the ROC and DCU Commander, DCU Commander and DCU Motor, and DCU Commander and DCU Rudder. The DCU Motor sends the motor status to the NMEA2000 network, and the Furuno autopilot sends the rudder status to the NMEA2000 network. The NMEA2000 to UDP device captures NMEA2000 messages and sends them to the ROC.

When the remote control system is in control, the DCU Motor disengages the actual throttle input and acts as a throttle input, and the DCU Rudder starts acting as a remote controller for the Furuno autopilot. The ROC has priority over autonomous operation through the ROS2 Network, which can be overridden with either manual or remote operation.

The DCU includes several fail-safe functions. A heartbeat signal must be sent by the commanding unit at set intervals to maintain operational mode. The vessel's local operator

can also revert to manual operating mode by moving the engine control lever from idle or pressing an emergency button.

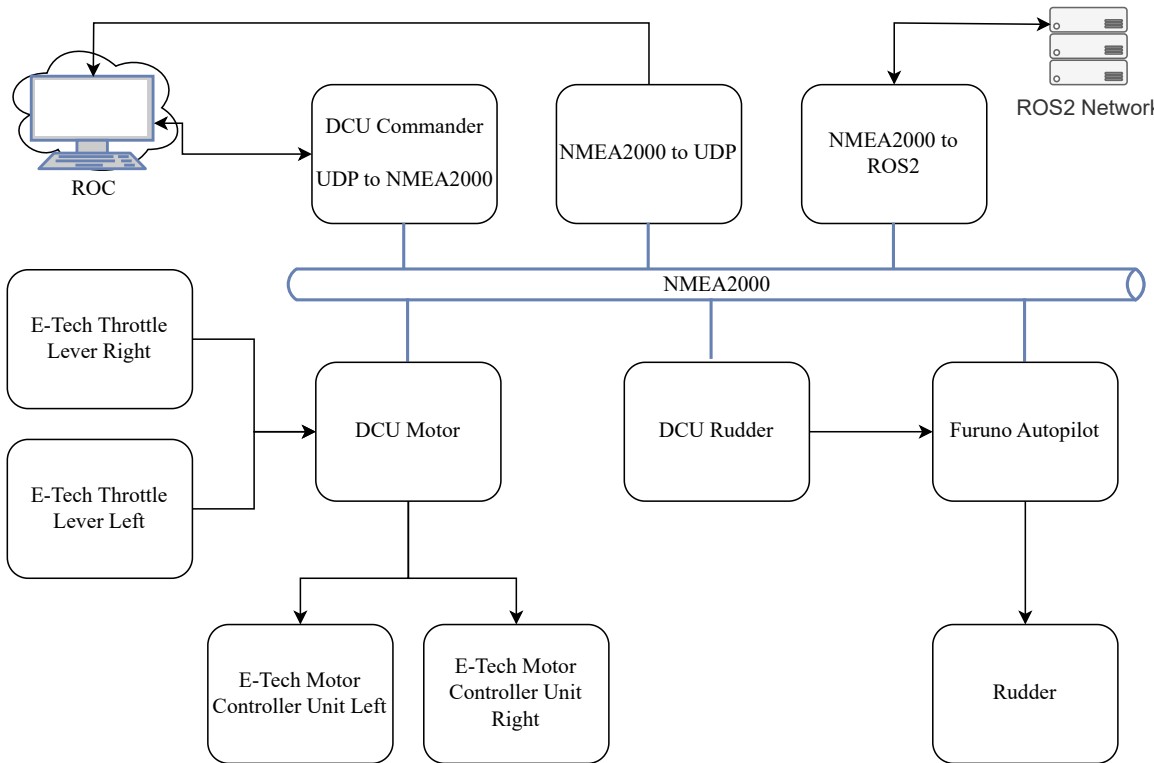

**Figure 10.** USV dynamic controller unit architecture.

## 4. Collecting and Applying Multi-Modal Sensor Data for Deep Learning in Maritime Environments

In the context of USVs, deep-learning-based autonomous navigation, guidance, and control (NGC) systems play a pivotal role in enabling the vessel to fulfill its mission without human intervention [35]. These systems perform tasks such as path planning, obstacle avoidance, and decision-making, all of which heavily rely on situational awareness from the USVs surroundings.

Deep learning algorithms not only facilitate the acquisition of situational awareness but also serve a dual purpose in this context. Firstly, they act as a decision support system for humans, whether they are stationed on board a USV or operating from an ROC. More importantly, these algorithms are instrumental in providing autonomous NGC systems with the necessary situational awareness. This awareness is crucial for the NGC algorithms to carry out their tasks both effectively and safely.

The process of acquiring situational awareness is streamlined by the use of deep learning and computer vision methods. These techniques are particularly useful for tasks such as detecting and tracking objects in maritime environments [36]. However, object detection in these environments presents a significant challenge due to factors such as varying light conditions, view distances, weather conditions, and sea waves. These factors can lead to false positive and false negative detections due to light reflection, camera motion, and illumination changes.

To overcome these challenges and ensure reliable situational awareness, a variety of sensors needs to be employed. Multi-modal sensor data are integral to the successful operation of the deep learning algorithms and, by extension, the autonomous NGC systems of the USV. In the following, we briefly discuss the modalities in our data collection setup.

Red, green, and blue (RGB) cameras capture color information and enable object recognition based on the object's color and shape. RGB cameras are crucial for situational

awareness, where visual observation of the surroundings plays a significant role. The RGB cameras also aid in mapping and localization by differentiating between various surfaces visible from the water level. Using RGB cameras in stereo vision configuration allows us further to capture depth information and enhance a USV's spatial awareness and object recognition capabilities. The depth perception is critical for tasks such as three-dimensional maritime mapping and navigating effectively by recognizing the spatial arrangement of maritime elements.

Thermal cameras capture heat signatures and are crucial for navigating in conditions with compromised visibility like fog or darkness. They detect heat-emitting objects and identify temperature variations in the sea surface, enhancing vehicle autonomy and reliability. In addition to situational awareness, they are useful in tasks like search and rescue missions and security measures. LiDAR sensors create detailed 3D point clouds, which are essential for environmental mapping and navigation in the domain of USVs. A significant issue with LiDARs is that the cloud presentations are difficult to reconstruct in poor weather conditions.

However, no single sensor is able to guarantee sufficient reliability or accuracy in all different situations. Sensor fusion can address this challenge by combining data from different sensors and by providing complementary information about the surrounding environment. Previous work has concentrated on sensor fusion methods for visible light and thermal cameras [37–39], as such cameras are cheap, and the data annotation is relatively easy compared to LiDAR point clouds [40]. Even though the visible light cameras have very high resolutions, poor weather and illumination changes can easily distort the image they produce. To tackle this, their data can be fused with data from thermal cameras. Even with the state-of-the-art methods for visible light and thermal camera sensor fusion, it is still difficult to detect very small objects. LiDAR point clouds do not contain information on the object's color or fine surface details, but their data can still be effectively used for object detection using neural networks [41]. The depth information from LiDAR produces a better understanding of the environment of the vessel when fused with data from other sensors.

Enhancing autonomous features in maritime environments necessitates the development of distinct deep learning applications tailored to their specific settings. While some challenges overlap with those faced by other autonomous vehicles, unique hurdles arise from dynamic and unpredictable environments, including open seas, ports, and congested waterways. Another significant challenge emerges from the inherent limitations and delays in the navigational responses of boats and ships when abruptly altering routes or directions. Unlike autonomous cars, these limitations compel researchers to design algorithms with an extended range for object detection. Moreover, the maritime domain presents a scarcity of research in comparison to other autonomous fields.

Securing adequate and relevant data for deep learning algorithms remains a primary challenge in the domain of autonomous maritime vessels. The scarcity and small scale of publicly available datasets in the maritime domain present significant obstacles in the development of deep learning algorithms for situational awareness in maritime environments. A comprehensive survey of available maritime vision datasets compares datasets in terms of data type, environment, ground authenticity, and applicable research directions [42]. If they were to be used with our USV, datasets from the Singapore Maritime Dataset [43] and SeaShips [44] differ substantially from the research environment in the Southwest of Finland. While the Åboships [45] dataset bears some similarity, its size and quality are inadequate. Also, these datasets are acquired using RGB cameras, and notably, no complementing datasets from other sensor types, such as stereo vision cameras, thermal cameras, or LiDAR, are currently available. Having access to multi-modal datasets is crucial for advancing autonomous capabilities in maritime applications to enhance the ability of deep learning algorithms to work in such environments in varying weather and time conditions.

In this section, we introduce the multi-modal sensing and data collection system integrated into our USV eM/S Salama. We aim to create synchronized and annotated multi-modal datasets to be published in open access. The first version of the sensing system

includes RGB and thermal cameras, a stereo vision camera, and a LiDAR. Radar and other sensors will be added to the setup later on. We also present a data platformthat stores the multi-modal sensor data in a format that is readily usable for the development of deep learning algorithms. We introduce the data we already have collected, along with a brief description of zero-shot labeling and synthetic data generation methods. The section also describes examples of deep learning algorithms we have developed to provide situational awareness for autonomous navigation algorithms.

## 4.1. Multi-Modal Sensing and Data Collection System

The eM/S Salama sensor data collection setup, shown in Figure 11, comprises three distinct RGB cameras oriented in different directions, three thermal cameras aligned with the RGB cameras, and two stereo vision cameras positioned with a central distance of 150 cm. A LiDAR sensor, IMU, and GPS antenna are also incorporated into the configuration. This setup enables the collection of diverse data types, enabling us to obtain situational awareness from the vessel's environment using deep learning and computer vision.

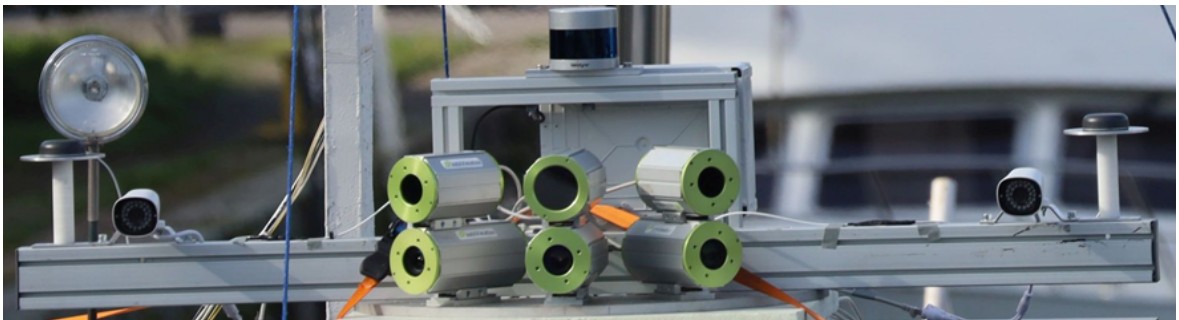

**Figure 11.** Sensoring setup at eM/S Salama.

The camera array consists of three weather-proofed RGB and thermal cameras, creating an approximately 140-degree panoramic view for daylight and thermal imaging. For high-resolution RGB imaging needs, we rely on the Allied Vision Prosilica GT 1930 [46] cameras. For thermal imaging, the Teledyne Dalsa Calibir 640 [47] cameras offer high sensitivity and a high 640 × 480 resolution.

Complementing our sensor array is a compact 360-degree Velodyne VLP-32C ultra-puck LiDAR [48]. This LiDAR sensor generates a point cloud 3D map of the surroundings of the vessel. The GPS coordinates and IMU data are also recorded, allowing us to know the location and the pose of the vessel.

The sensor data collection system, as shown in Figure 12, consists of the RGB and thermal cameras, the LiDAR, and software services. The data handler services are responsible for acquiring the raw data from all of the sensor sources and preprocess the data to enable further processing and data storage later in the pipeline.

Synchronized sensor data are needed for multi-modal sensor fusion. The process orchestrator service is responsible for launching and synchronizing the data collection between the handler services. A process orchestrator is also used to command the handler services in situations where we need to change sensor configurations on the fly. The software services use Message Queuing Telemetry Transport (MQTT) protocol over a Mosquitto message broker for communication between each other. The software services are deployed in separate Docker containers to enable a scalable, portable, and modular system that complies well with a microservice architecture.

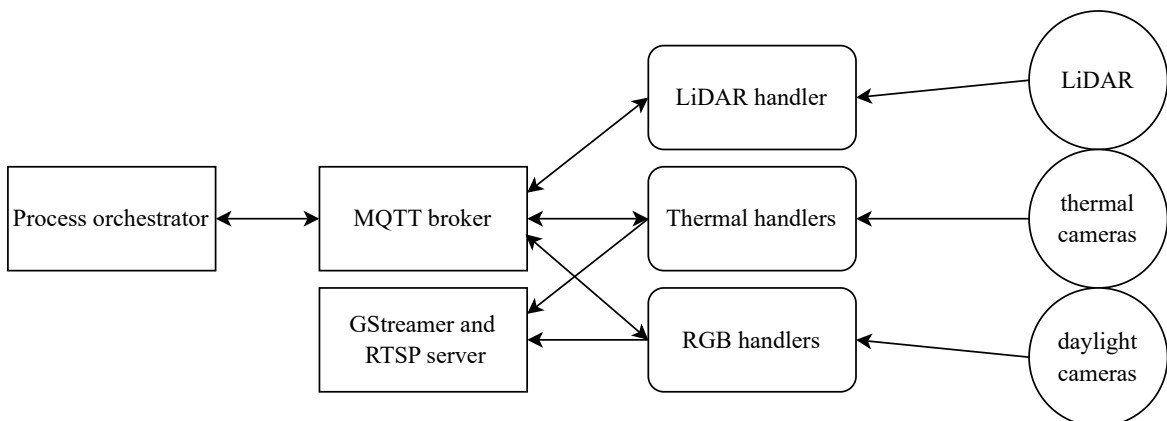

**Figure 12.** Sensor data collection system architecture.

The raw data from RGB and thermal cameras is encoded with GStreamer and published as RTSP streams. The LiDAR data are stored in Packet Capture (PCAP) [49] files, which include timestamps that enable the synchronization with camera data.

Design and Implementation of the On-Shore Data platform

Local data storage capacity on eM/S Salama is limited. In addition, developing deep learning models requires considerable computing resources. To solve this challenge, we have developed an on-shore data platform that is designed to store the multi-modal data collected on eM/S Salama. The data platform can also be used to enrich the data with information collected from external sources. This includes, for example, accurate weather data and AIS data concerning the other nearby maritime vessels.

Because of the multi-modal nature of the sensor data that need to be stored in the system, we chose a hybrid data storage approach, where we combine different types of databases best fit for different data formats and use cases. Our two main components are a PostgreSQL relational database that is used to index all our data objects and their related metadata and an Ambry object storage, which we can use to store all binary data types.

We utilize PostgreSQL to store AIS data, along with data object metadata. The Structured Query Language (SQL) serves as the standard language for database creation and manipulation, and PostgreSQL adheres to SQL conventions as outlined in the documentation [50]. As a mature and open-source object-relational database, PostgreSQL offers high performance and reliability. As we progress with the deep learning components and other sensor data, the relational database will be used to store other structured data types.

For unstructured data, we needed object storage. We chose LinkedIn's Ambry for this purpose. It implements a REST front-end and a non-blocking I/O back-end. This provides us with high-performing way to store immutable data that can be accessed via a HTTP API.

The data platform is deployed on the Turku UAS on-shore Proxmox-based cloud. This provides us with an added layer of security, as the virtual machines on the platform are isolated from the Internet by default. Only the machines where the gateway API services are run are reachable from outside Proxmox.

Within Proxmox, we run a microservice architecture, as shown in Figure 13. The data source gateway is the entry point for data into the system. The gateway consists of multiple interfaces capable of ingesting data in different formats delivered via different protocols. The first implemented protocols were MQTT and HTTPs. We use these to ingest both batch and streaming data.

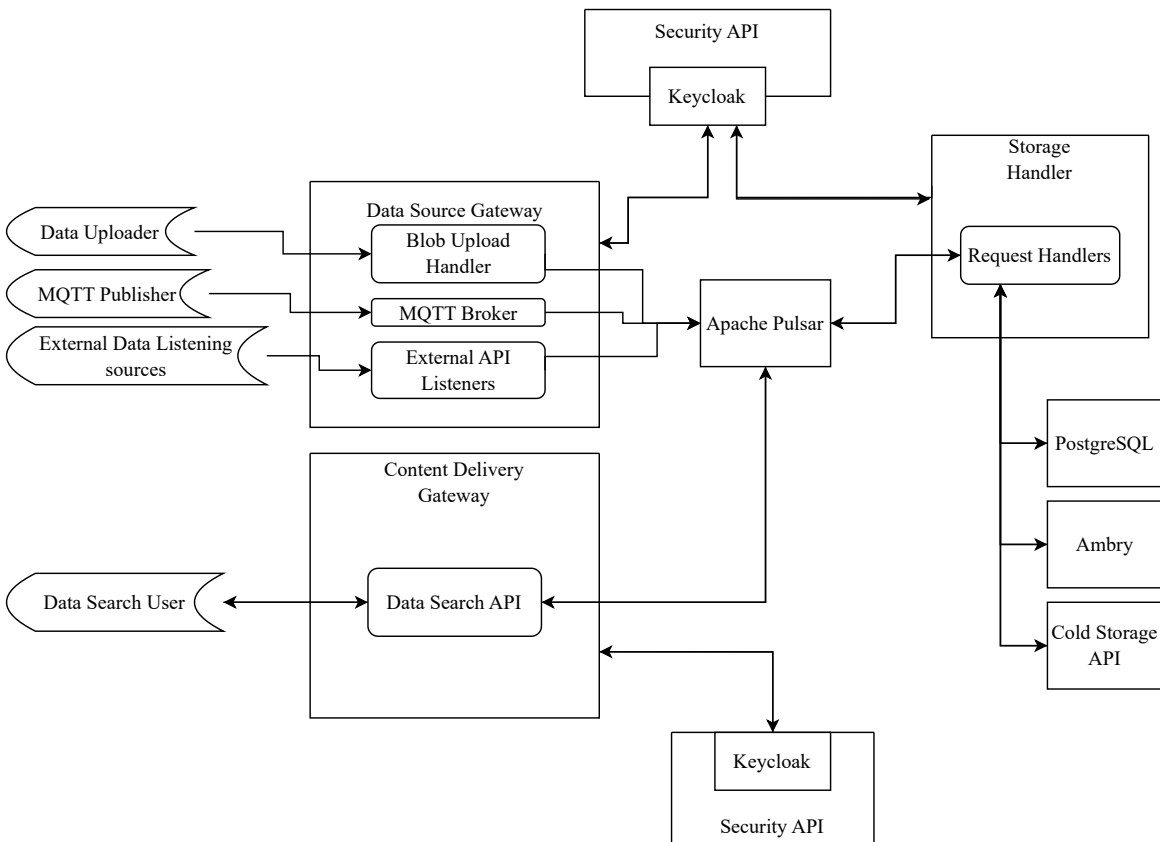

**Figure 13.** Data platform services.

For communication between the microservices and to deliver data from the gateway to the data storage handlers, we use Apache Pulsar. Pulsar's publish–subscribe pattern meets our requirements for performance, durability and multi-tenancy. We use Keycloak to identify and access management throughout our system, the Pulsar endpoints being the key points.

The storage handler services make sure that all data acquired from the gateway are associated with metadata and first stored in the local hot storage that Ambry and PostgreSQL form and later moved onto cold storage. The metadata related to all data objects consist of fields describing the data object and where it is currently stored. This enables the data search queries to only target the metadata.

### 4.2. Data for Deep Learning

In the summer of 2022, we established our first data collection framework and recorded a total of 72 h of video over four days using our stereo vision camera and a LiDAR. Following a thorough exploration of our data, we initiated the data annotation process using MATLAB R2023a, covering 120,216 RGB images, 53,108 stereo images, 60,108 multi-view images, and 36 h of LiDAR data. Subsequently, in 2023, we expanded our setup by integrating three additional RGB cameras and three thermal cameras. This enhanced configuration allowed us to gather over 93 h of video, with the data annotation process currently in progress. We mainly collected data during the summer period because the sea is frozen during the winter in Finland.

Samples from our collected data are shown in Figure 14.

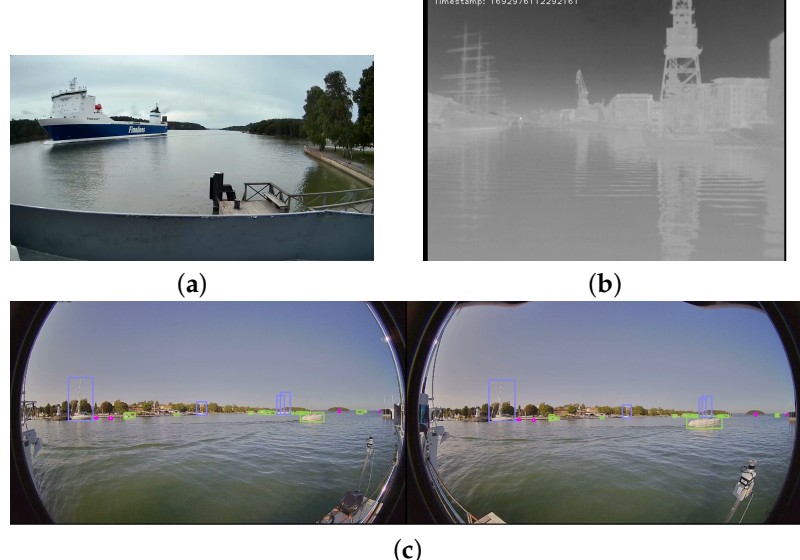

**Figure 14.** Samples of our dataset from summer 2023. (**a**) Captured image from RGB camera. (**b**) Captured image from thermal camera. Cold objects are black and hot objects white. Scales of gray indicate variations between these two. (**c**) Captured and annotated image from stereo vision cameras.

### 4.2.1. Zero-Shot Labeling

We are currently developing an innovative framework aimed at mitigating the labor-intensive task of annotating extensive datasets by utilizing the capabilities of zero-shot object detection and object tracking. Given the wealth of data accumulated from diverse sensors over numerous hours, the traditional approach of manual annotation becomes impractical and time-intensive. Leveraging state-of-the-art techniques in zero-shot object detection allows the system to generalize and identify objects even without specific annotations, streamlining the annotation process significantly.

Additionally, our framework incorporates robust object-tracking mechanisms, ensuring the continuity of object identification across frames. By integrating these advanced technologies, we aim to enhance the efficiency of dataset annotation, making it more feasible and scalable for large datasets generated from multi-sensor environments [51].

### 4.2.2. Synthetic Data Generation with GANs

Generating synthetic data that mimics, for instance, various weather and time of day conditions could be more efficient than actually obtaining real data in all possible time and weather combinations. While synthetic images can be generated manually [52], automated tools make the process much faster and more straightforward. For this purpose, we propose a concept to generate synthetic data using Generative Adversarial Networks (GANs) [53].

GANs are a class of deep learning algorithms, which can generate new data instances that could pass for real data. This makes GANs particularly useful for tasks such as ours, where we need to generate synthetic data that mimic various environmental conditions.

Creating a diverse dataset that includes novel, realistic scenarios is crucial for enhancing the robustness of deep learning models. In maritime environments, where weather conditions can vary significantly, it is essential for our models to detect objects under different circumstances. To address this need, the use of synthetic data is vital. By creating synthetic datasets that include a wide range of environmental conditions, we can expose the model to various situations, ensuring its effective performance under different scenarios. This not only aids the model in better adaptation but also fortifies it against unexpected challenges. It is a key step in constructing a comprehensive and reliable object detection system for maritime applications. In the future, we plan to use Transformer algorithms [54,55] to solve for example the class imbalance issues by adding more objects of each category.

*4.3. Enhancing Maritime Safety with Situational Awareness from Deep Learning Algorithms*

This section explores the application of deep learning algorithms to enhance maritime safety. This is a critical aspect of decision support and autonomous operations in USVs. Our research focuses on refining these algorithms to improve their efficiency and integration into existing systems, thereby providing crucial situational awareness for both human operators and autonomous navigation algorithms.

We explore three key areas of deep learning that have been applied to our collected data, each playing a pivotal role in maritime safety:

- Object detection in maritime environments: This involves the use of deep learning algorithms to identify and classify objects in maritime settings, which is crucial for avoiding obstacles and ensuring safe navigation.
- Object tracking in maritime environments: Once objects are detected, it is essential to track their movements accurately. This allows for the better prediction of potential collisions and assists in decision-making for course adjustments.
- Horizon line detection in maritime environments: The ability to identify the horizon line is vital for maintaining orientation and stability, especially in rough sea conditions. It also aids in the calibration of other detection and tracking algorithms.

Subsequent sections present a detailed examination of each research area. The overarching objective is to augment maritime operation safety and efficiency via the practical implementation of these refined deep learning algorithms.

4.3.1. Object Detection in Maritime Environments

Our image datasets can benefit from the High-Resolution Daytime Translation (HiDT) model [56], which enhances dataset diversity and performance. HiDT generates synthetic data across various time-of-day scenarios. Notably, HiDT can simulate eighteen different weather conditions. In this section, we demonstrate that additional synthetic images added to a dataset can help in achieving better results in object detection.

In our experiments, we utilize and release the Turku UAS DeepSeaSalama-GAN dataset 1 (TDSS-G1) [57]. Assembled in the Southwest Finnish archipelago area at Taalintehdas, this dataset employed two stationary RGB cameras to collect data in August 2022. TDSS-G1 comprises 199 original images and a substantial addition of 3582 synthetic images, resulting in a total of 3781 annotated images. Table 3 shows the number of images in the training, validation, and test sets. The number of images in a dataset and their diversity in terms of, for example, different viewing angles, lighting conditions, occlusions, scales, and hull parts is very important for the performance of object detection models [58]. It has to be noted that the dataset published here is relatively small and mainly used to showcase how the performance of object detection can be improved by generating additional synthetic images to train the models.

**Table 3.** The number of images in the TDSS-G1 dataset.

| Dataset | Train Set | Valid. Set | Test Set |
|---|---|---|---|
| Real | 199 | 49 | 50 |
| Real + Synthetic | 3781 | 49 | 50 |

These images feature high-quality annotations of maritime objects, categorized into three classes: motorboats, sailing boats, and seamarks. The distribution of labels within TDSS-G1 is as follows: motorboats (62.1%), sailing boats (16.8%), and seamarks (21.1%). The number of instances in the original images is shown in Figure 15.

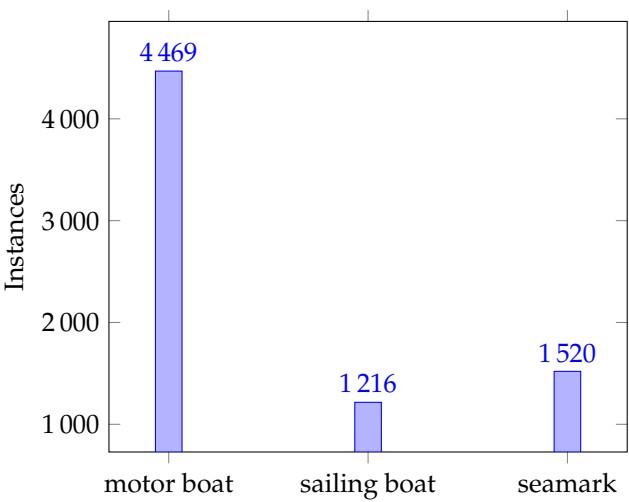

**Figure 15.** Dataset class instances for real data.

Figure 16 showcases a selection of images produced by HiDT, effectively demonstrating its efficacy with our datasets. For a more detailed description of the dataset, refer to Zenodo [57]. In the future, we can further enhance its image-generation ability by collecting and utilizing additional training samples from specific scenarios, such as nighttime maritime scenarios.

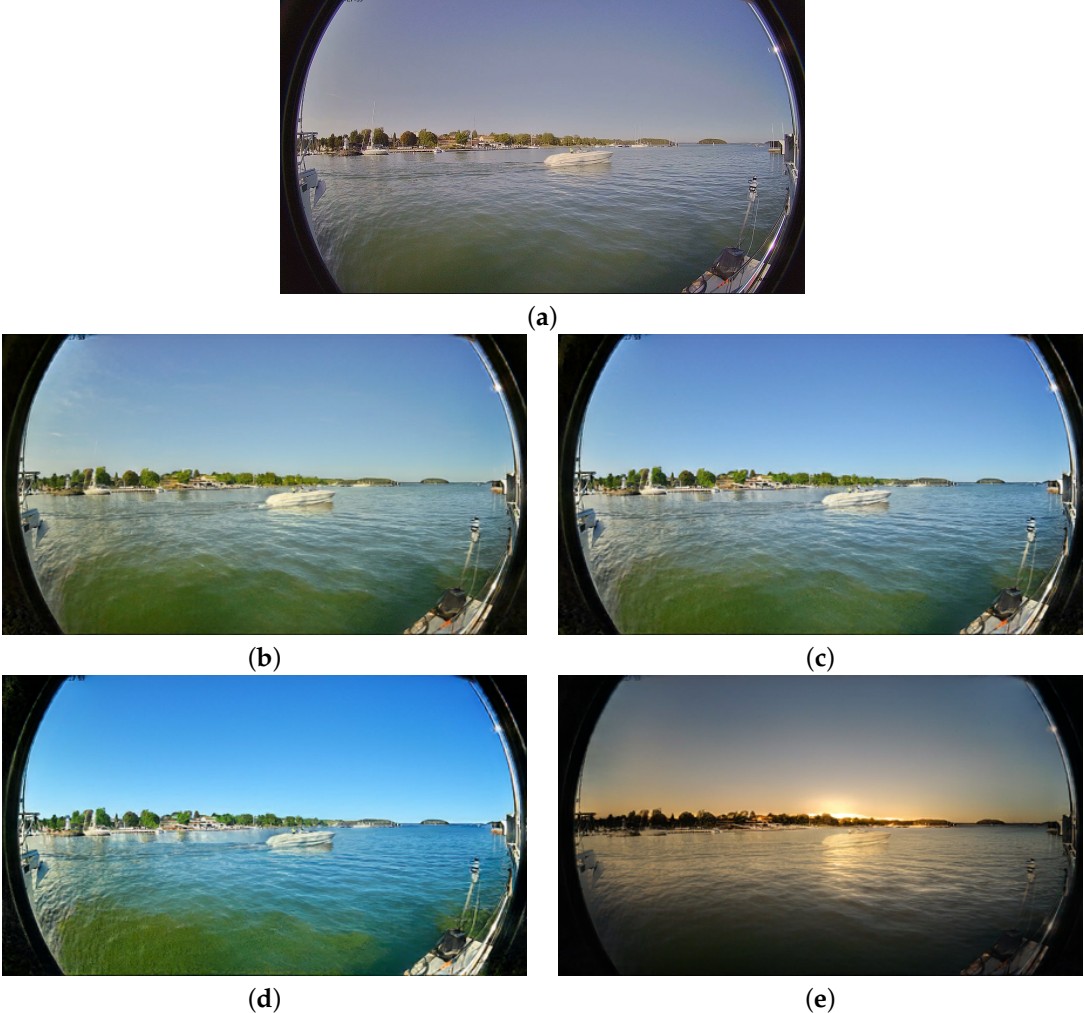

**Figure 16.** *Cont.*

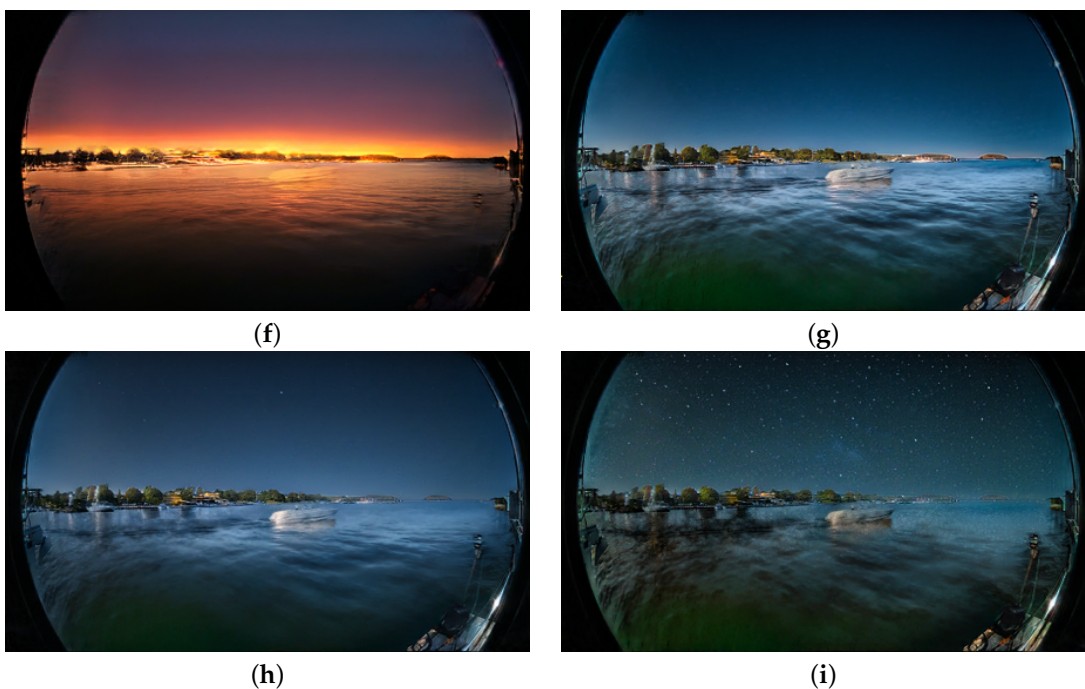

**Figure 16.** Selected sample from our synthetic image dataset TDSS-G1: side by side, (**a**) original image, (**b**–**d**) three day images, (**e**,**f**) two sunset images, and (**g**–**i**) three night images.

In our object detection experiments [59], we utilize the YOLOv7 [60] model and conduct an extensive analysis comparing YOLO results with and without the incorporation of synthetic data. As detailed in Table 4, the inclusion of synthetic data leads to a significant enhancement in the mean Average Precision (mAP), reaching an impressive mAP of 0.822. In the maritime environment, detecting small objects is a critical challenge. To address this issue, it is essential to evaluate deep learning models also for different sizes of objects [61]. The results for the evaluation of small, medium, and large objects in terms of size in pixels can be seen in Table 5. We can note that the performance increases significantly when the size of the object increases.

**Table 4.** mAP@0.5 results with YOLOv7 model trained and tested with real and synthetic data.

| Train Set | Test Set | mAP@0.5 |
|---|---|---|
| Real | Real | 0.788 |
| Real + Synthetic | Real | 0.822 |
| Real | Real + Synthetic | 0.518 |
| Real + Synthetic | Real + Synthetic | 0.774 |

**Table 5.** mAP@0.50:0.95 results with the YOLOv7 model trained on real + synthetic data and tested with real data for different object sizes. Note that the sizes are in pixels.

| Average Precision Results | mAP@0.50:0.95 |
|---|---|
| Small (size < 32∗32) | 0.145 |
| Medium (32∗32 < size < 96∗96) | 0.509 |
| Large (size > 96∗96) | 0.917 |

mAP is a metric to evaluate the performance of object detection models by averaging precision over recall and classes. Precision is true positives over predicted positives, and recall is true positives over actual positives. mAP@0.5 means that the intersection over union (IoU) threshold for determining true positives is set to 0.5. IoU is the ratio of the area of overlap between the predicted bounding box and the ground truth bounding box to

the area of their union. Setting the IoU threshold to 0.5 means that only predictions that have at least 50% overlap with the ground truth are considered true positives. A higher mAP@0.5 value indicates a better performance in object detection.

Figure 17 illustrates the precision-recall curves, which show that models trained on the combined dataset outperform those trained solely on real data. Due to the static nature of the images, the seamarks present in them are always detected perfectly. This underscores the value of integrating synthetic data to strengthen the object detection capabilities of the models, thereby affirming the importance of our approach in achieving robust and adaptable performance across diverse environmental conditions.

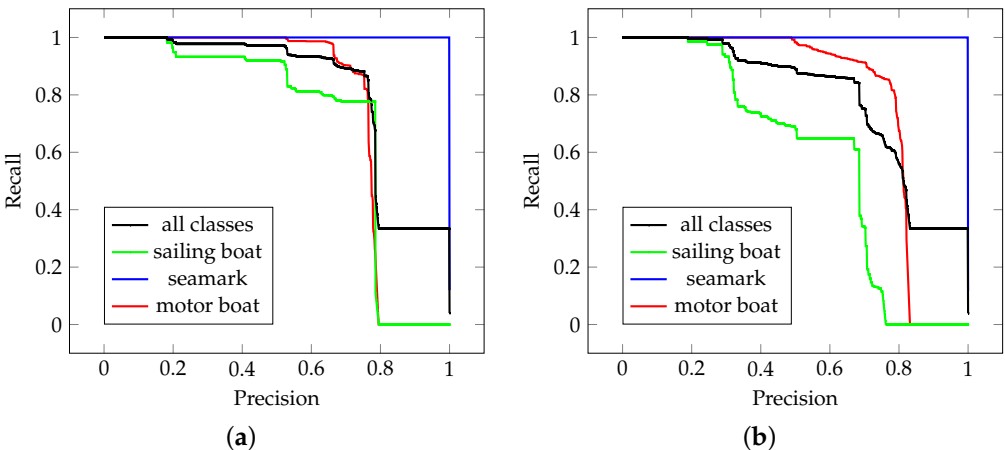

**Figure 17.** Precision-Recall curve for YOLOV7 model (**a**) trained on real data + synthetic data, (**b**) trained on real data.

### 4.3.2. Object Tracking in Maritime Environments

Object tracking is a crucial aspect of maritime safety, as it allows for the accurate monitoring of detected objects' movements. This is essential for predicting potential collisions and assisting in decision-making for course adjustments.

Various approaches are employed by object-tracking algorithms to address the challenge of tracking object in different environments. Among these, the DeepSORT model [62] stands out as one of the most efficient and accurate options in its class. DeepSORT uses a deep learning model to generate appearance features for bounding box regions identified by an object detection model. These features are then transformed into a cost matrix, with its components representing cosine distances to the appearance features stored in the track, facilitating data association.

The data association process employs the matching cascade algorithm, where matching with detections occurs sequentially, starting with the most recently updated tracks. Tracks that correspond with detections through data association store the corresponding detection's appearance features and state estimates updated by the Kalman filter.

Figure 18 illustrates the proposed object tracking architecture. Based on our proposed architecture, to optimize DeepSORT's performance in maritime environments, we made slight adjustments. Initially, our custom YOLOv8 [63] model was employed. Subsequently, we incorporated a lightweight MobileNet algorithm [64] for the tracking component. This hybrid approach ensures effective object detection and swift tracking, a critical requirement for real-time decision-making in maritime scenarios.

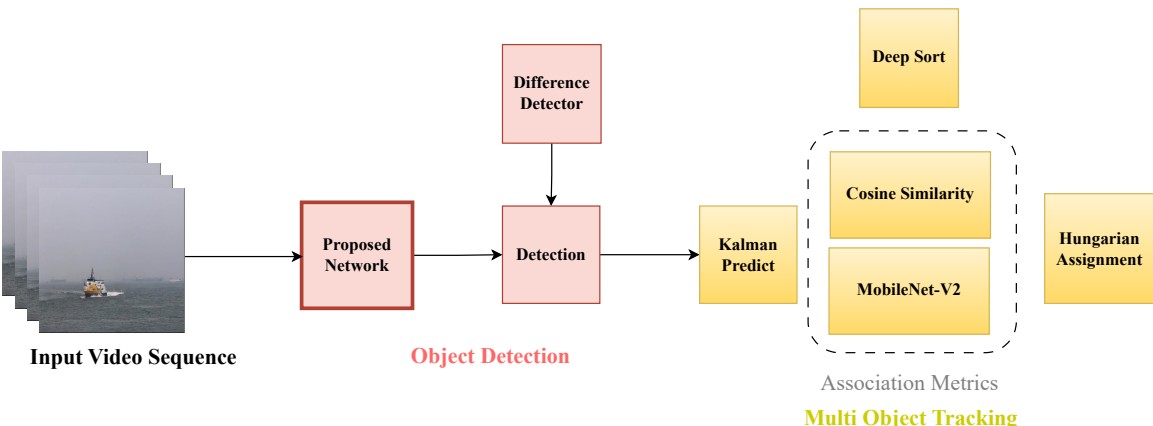

**Figure 18.** Proposed network architecture for object tracking.

Despite these customizations, challenges persist in maritime environments due to the instability of images captured by cameras on boats or ships. This variability poses a challenge for the DeepSORT algorithm's Kalman filter component in maintaining object tracking. To address this issue, we have opted to tackle the problem with horizon line detection, which is detailed in the subsequent section. This approach not only improves object tracking but also enhances the overall situational awareness by providing a more stable and accurate view of the maritime environment.

Figure 19 illustrates an example of our object tracking method applied to a video sourced from the Singapore Maritime Dataset. This dataset comprises 81 existing and freely accessible videos related to maritime scenarios [65]. These scenes often exhibit a tolerated sight, resulting in significant variations in the angles from the horizon line. In the figure, we can see that the algorithm has detected three different ships whose movement it is continuously tracking.

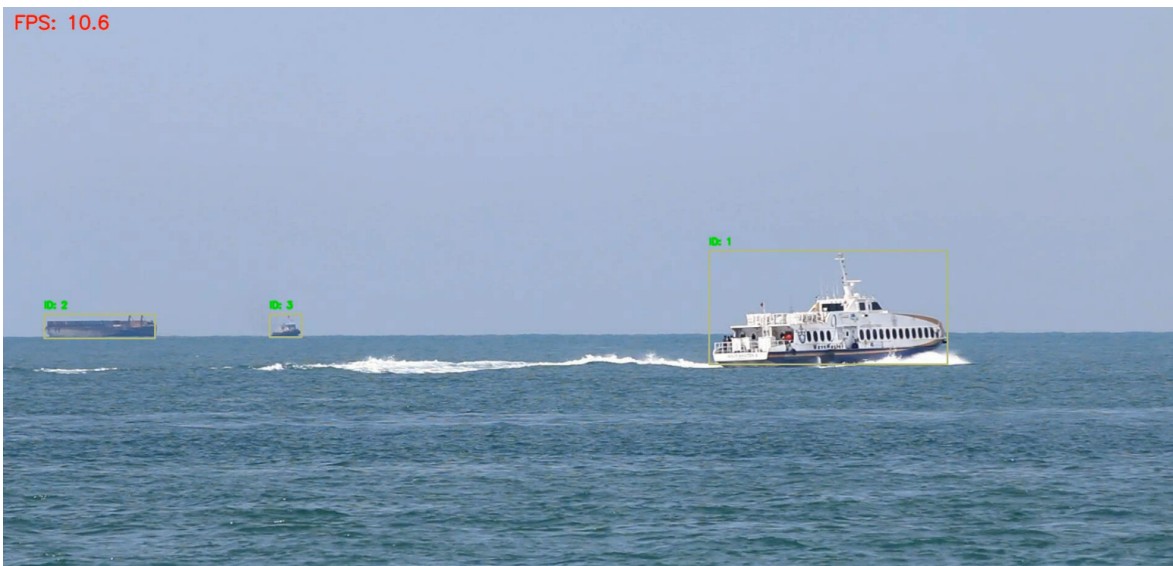

**Figure 19.** Object tracking applied to data from Singapore Maritime dataset [65]. The yellow squares show detected objects that the algorithm is tracking and the green texts show a unique ID given to each object.

### 4.3.3. Horizon Line Detection in Maritime Environments

Semantic line detection, particularly the identification of the horizon line (sea line), is crucial in maritime environments. It influences various decision-making processes and aids in the calibration of other detection and tracking algorithms.

Our research leverages a state-of-the-art algorithm as the baseline, which we subsequently tailor and optimize specifically for maritime settings. The foundational paper guiding this optimization is titled "Deep Hough Transform for Semantic Line Detection" [66]. The architecture of this algorithm is detailed in Figure 20.

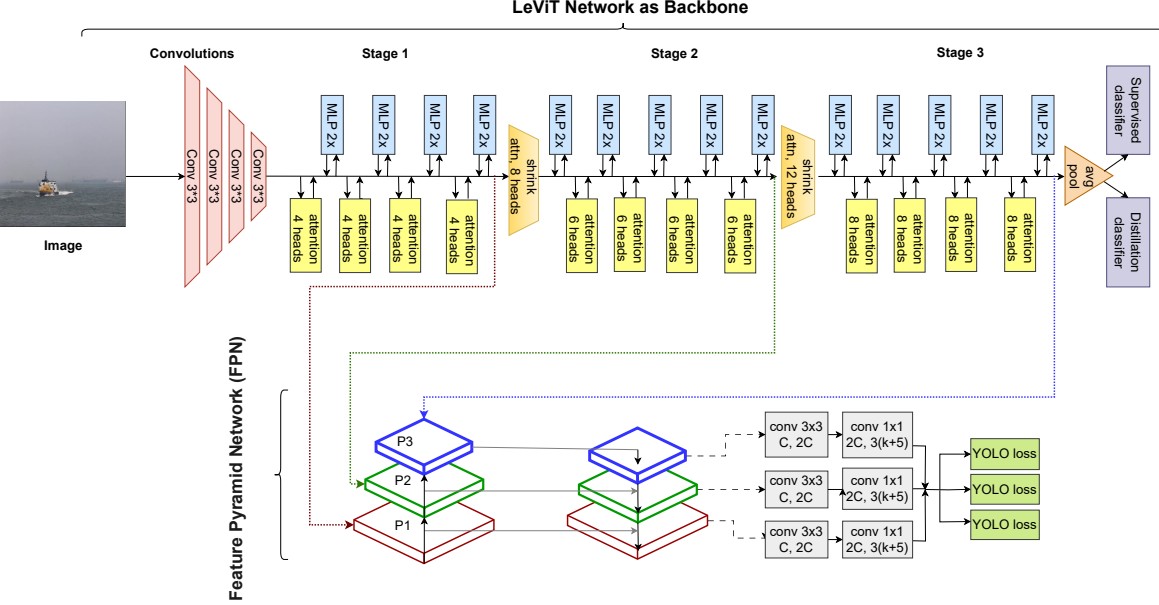

**Figure 20.** Semantic line detection network architecture proposed in Deep Hough Transform for Semantic Line Detection. Adapted from [66].

The semantic line detection network consists of three main components: a backbone network, a Hough transform module, and a line detection head. The backbone network is a convolutional neural network that extracts feature maps from the input image. The Hough transform module performs a Deep Hough Transform (DHT) that aggregates the features along candidate lines on the feature map plane and assigns them to corresponding locations in the parametric space, where each location represents a line with a specific slope and bias.

The algorithm in Figure 21 is employed to identify potential lines in the parametric space. A line refinement network is then utilized to prioritize and select the most representative horizon line among them. The line detection head is a fully connected layer that predicts the confidence scores and offsets for each location in the parametric space. The refinement network is powered by a Vision Transformer (ViT), introduced in the paper "An Image is Worth 16 × 16 Words: Transformers for Image Recognition at Scale" [54].

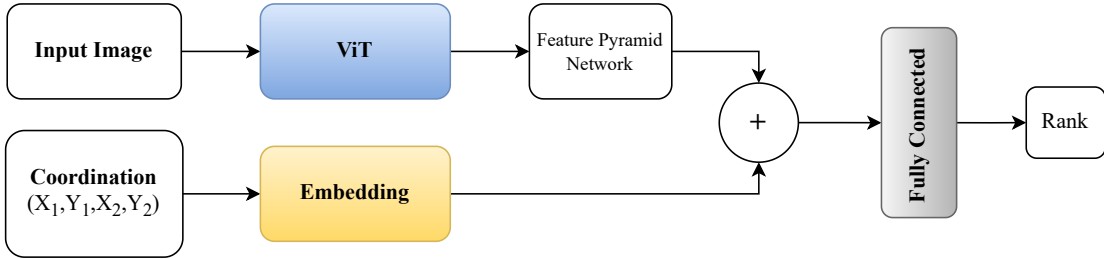

**Figure 21.** Proposed network architecture for horizon line detection.

Additionally, we incorporate a Feature Pyramid Network to extract a diverse set of features from the backbone network. The final output is a set of semantic lines that have

high confidence scores and refined parameters. The detected lines in the parametric space are then converted back to the image space by a Reverse Hough Transform (RHT).

The coordinates, presented as a tuple $(x_1, y_1, x_2, y_2)$, are also input into an embedding layer. Ultimately, a fully connected network assesses the ranking of these coordinates, determining the horizon line in the given input.

To evaluate our method, we have tested it on the Singapore maritime dataset and our dataset from the Finnish archipelago. Figure 22 illustrates the output of our method.

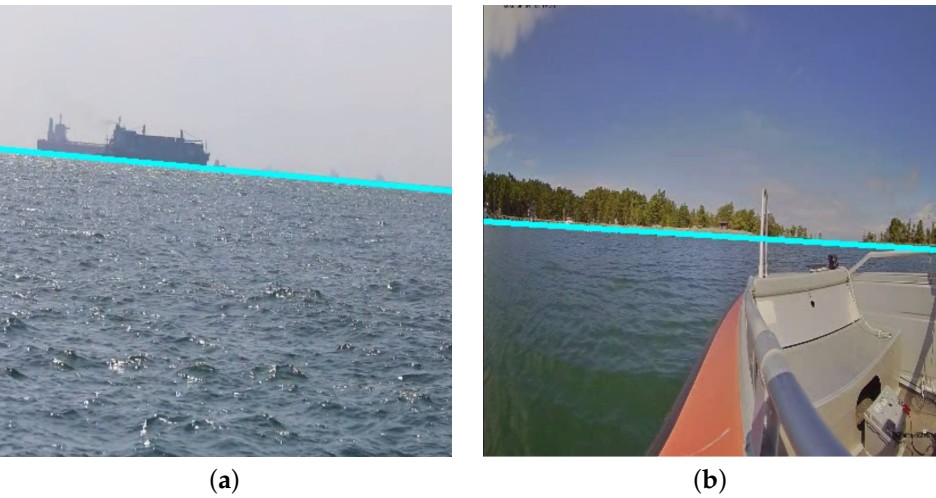

(**a**)          (**b**)

**Figure 22.** Sample results from the horizon line detection. The blue line is the detected horizon line. (**a**) Singapore Maritime dataset [65]. (**b**) Finnish archipelago.

The horizon line detection is not only crucial for maintaining orientation and stability, especially in rough sea conditions, but it also aids in the calibration of other detection and tracking algorithms. By accurately detecting the horizon line, we can provide a more stable and accurate view of the maritime environment, thereby improving object tracking and enhancing overall situational awareness.

## 5. Discussion

Our journey in developing our USV, eM/S Salama, commenced in 2022. We have now reached a point where the USV can be operated both manually and remotely. The substantial size of the vessel equips us with the ability to operate under challenging weather conditions and to gather data from a variety of weather scenarios. For instance, we can collect data under heavy fog conditions and develop deep learning algorithms to gain relevant situational awareness, even when human vision navigation is not feasible.

USVs necessitate precise situational awareness to ensure safe decision-making and effective collision avoidance. Our USV utilizes a multi-modal sensing system, including RGB cameras, thermal cameras, and LiDAR sensors. We have collected and annotated data, enabling us to leverage them in deep learning algorithms and gain situational awareness of the vessel's environment. Ensuring accurate sensor measurements is critical for reliable operation. Incorrect sensor and camera calibration [67] can lead to navigation errors, degrade performance, and increase the risk for collision. We aim to develop sensor autocalibration and registration methods so that the sensors can self-adjust and align multiple sensor's data automatically.

We need to gather a comprehensive dataset to enable precise vision-based trajectory-tracking control and localization estimation for our USV [68,69]. Image data capturing different lighting conditions and times of day are crucial for building a robust object detection model [58]. To ensure our dataset represents diverse weather conditions, this paper showcases the benefits of integrating synthetic data into model training. As outlined in Table 4, including synthetic data notably boosts the mean average precision (mAP@0.5),

improving it from 0.788 to an impressive 0.822. This underscores the significance of incorporating synthetic data to enhance model performance across various weather conditions.

We currently leverage the obtained situational awareness information in the development of autonomous NGC for our USV. This involves the creation of collision avoidance methods in compliance with COLREG rules. COLREG compliance is a very challenging task due to its ambiguous nature, which can lead to many different interpretations of the rules [70]. The autonomous NGC system will be built on the ROS2 navigation stack, which facilitates safe and efficient movement. The integration of RGB cameras, thermal cameras, and LiDAR sensors for Simultaneous Localization and Mapping (SLAM) will serve the fundamental purpose of creating a local map for precise navigation. RGB cameras play a pivotal role in capturing visual information, aiding in feature extraction and visual odometry to build a visual representation of the environment. Thermal cameras further contribute by providing critical data that enhance navigation capabilities, especially in challenging conditions where RGB cameras might face limitations. Meanwhile, LiDAR sensors provide depth information, facilitating the creation of a comprehensive and accurate map. The fusion of the data from these sensors in a SLAM framework enables the system to construct and continuously update a local map, allowing our USV to autonomously navigate through its surroundings with increased accuracy and adaptability. This approach leverages the strengths of each sensor modality, overcoming individual limitations and providing a robust foundation for effective and reliable autonomous NGC systems.

Looking ahead, we intend to augment the USV with integrated waterproof drones and underwater drones, thereby creating a heterogeneous swarm [71]. This swarm will enhance our understanding of the vessel's surroundings. Drones at higher altitudes can sense farther than the sensors on the vessel's roof, while underwater drones offer a novel perspective on underwater activities. The USV, equipped with substantial computing and power resources, will function as the swarm's master node, housing the primary ICT infrastructure. The drones, acting as slave nodes, will connect to the master node, receiving directives and supplying data to improve situational awareness. Leveraging satellite imagery could enhance the USV situational awareness even further [72].

The marine regulations also mandate observation based on sound. The COLREGs [18] specify several types of sound signals, which are used in different situations [73] and should thus be observed. We are currently building an array with eight microphones on the USV to be able to observe sounds and their directions. Audiosonic sensing could also be applied for object and event detection and localization [73–75]. Fusing sound sensor data with camera, LiDAR, and radar data would also offer improved reliability and accuracy for situational awareness in different conditions.

Research on navigation in Global Navigation Satellite System (GNSS)-denied environments is one of our current focus areas. Robust navigation is essential for safety when GNSS signals are being jammed or spoofed. Methods of navigation based on vision, LiDAR, and inertial sensors to maintain accurate position and trajectory need to be developed for situations where GNSS signals can not be used.

The future integration of X-band radar data for object detection in adverse conditions, such as darkness and fog, will enhance the robustness of our USV in the future. By leveraging the ability of radar to penetrate environmental obstacles and its resistance to interference from weather conditions, such as rain or fog, our system will maintain reliable object detection capabilities even in challenging scenarios. Advanced sensor fusion techniques, such as Kalman filtering or neural-network-based algorithms, will be used to combine radar data with inputs from other sensors to provide a holistic understanding of the surroundings.

Digitalization is currently transforming fairways and fairway services. Projects and authorities are exploring and piloting the potential of digital services, such as those based on the S-100 Universal Hydrographic Data Model [76], for delivering navigational warnings and aids to navigation. These digital services are particularly advantageous for USVs, ensuring they always have access to the most current navigational information. Among

these initiatives is the MaDaMe project [77], coordinated by Turku University of Applied Sciences, which is playing a pivotal role in this digital transformation.

Once the autonomous features of our USV have been further developed, it will open up a wide range of applications for the use of our USV across various sectors. Below, we list some potential use cases for the USV and the developed deep learning algorithms:

- Commercial : Our USV can serve as a sensor platform for detecting floating objects, which can be crucial for shipping companies to avoid collisions and ensure safe navigation. It can also monitor underwater objects, providing valuable data for industries such as offshore wind farms.
- Civil Security: The USV can be used for surveillance and patrolling tasks, providing an extra layer of security in ports, marinas, and coastal areas. It can also serve as a platform for deploying aerial [78] or underwater drones for more detailed inspections [79].
- Defence: In the defence sector, our USV can be used for reconnaissance missions, mine detection, and anti-submarine warfare. Its ability to operate autonomously reduces the risk to human operators and allows for operations in hostile environments.
- Environment: The USV can play a significant role in environmental monitoring. It can measure water quality parameters and detect hazardous gases, providing real-time data for environmental agencies and researchers. It can also aid in wildlife conservation efforts by tracking marine life and studying their habitats.
- Search and Rescue: In the future, our USV could be equipped with life-saving equipment and used in search-and-rescue operations. Its ability to navigate autonomously and cover large areas could prove invaluable in locating and assisting people in distress at sea.

In addition to the deep learning research, our USV serves as a versatile tool for various research applications. It plays a crucial role in investigating the compatibility of mobile and satellite networks for remote vessel operations and data transmission from onboard sensors. Additionally, the USV is instrumental in cybersecurity studies, which encompass the identification and rectification of vulnerabilities, risk management, and the formulation of safety protocols.

## 6. Conclusions

In this article, we detailed the development and application of a test platform comprising a USV and an ROC, designed to advance autonomous maritime operations. The USV and ROC serve as a robust platform for collecting multi-modal sensor data, crucial for developing deep learning algorithms that enhance situational awareness and facilitate autonomous navigation decisions.

We developed a multi-modal sensing and data collection system integrated into the USV eM/S Salama, which allows for sensor fusion to provide a comprehensive understanding of the environment surrounding the USV. We aim to contribute to the research community by gathering and annotating sensor data related to various research topics associated with the presented test platform. These data will be made available as open-access datasets, providing valuable resources for further research in the field of autonomous maritime operations.

The deep learning algorithms we develop provide decision support to human operators, both onboard and in the ROC. Our research also underscores the significant contribution of synthetic data, generated by the HiDT model, in enhancing the performance of object detection models. By integrating synthetic images into the TDSS-G1 dataset [57], we achieved a notable improvement in mAP, highlighting the potential of our approach in obtaining robust performance across diverse environmental conditions.

The onshore data platform, designed to store multi-modal data collected on eM/S Salama, ensures secure and efficient data management and facilitates the development of deep learning algorithms.

In conclusion, this article demonstrates the feasibility and potential of our test platform for applying deep learning and computer vision methods to advance autonomous maritime

operations. The challenges encountered and the solutions developed provide valuable insights for other researchers and practitioners in the field. Our forthcoming contribution to the scientific community will be a detailed description of the autonomous NGC system currently under development.

**Author Contributions:** Conceptualization, J.K. (Juha Kalliovaara), T.J., M.A., A.M., J.H., J.A., M.S., A.P., J.K. (Juho Koskinen), and R.M.M.; Funding acquisition, J.K. (Juha Kalliovaara) and J.P.; Investigation, J.K. (Juha Kalliovaara), T.J., M.A., A.M., J.H., M.S., A.P., J.K. (Juho Koskinen), T.T., and R.M.M.; Methodology, J.K. (Juha Kalliovaara), T.J., M.A., A.M., J.H., and J.A.; Project administration, J.K. (Juha Kalliovaara); Software, J.K. (Juho Koskinen) and T.T.; Supervision, J.K. (Juha Kalliovaara); Visualization, J.K. (Juha Kalliovaara), M.A., A.M., J.H., J.A., J.K. (Juho Koskinen), T.T. and R.M.M.; Writing—original draft, J.K. (Juha Kalliovaara), T.J., M.A., A.M., J.H., and J.A.; Writing—review & editing, J.P. All authors have read and agreed to the published version of the manuscript.

**Funding:** This research was co-funded by European Regional Development Fund, with the REACT EU support for regions to recover from the coronavirus pandemic, with grant numbers A78624 and A80845. This research was co-funded by the Finnish Ministry of Education and Culture in Applied Research Platform for Autonomous Systems (ARPA) project, 2020-2023. This research was co-funded by Business Finland in 5G-Advanced for Digitalization of Maritime Operations (ADMO) project.

**Data Availability Statement:** Data is contained within the article: The Turku UAS DeepSeaSalama—GAN dataset 1 (TDSS-G1) (1.0) is available in Zenodo [57].

**Conflicts of Interest:** The authors declare no conflicts of interest.

## Abbreviations

The following abbreviations are used in this manuscript:

| | |
|---|---|
| AAWA | Advanced Autonomous Waterborne Applications Initiative |
| Ah | ampere-hours |
| AIS | Automatic Identification System |
| BIB | Battery Interface Box |
| CAN | Controller Area Network |
| COLREG | Convention on the International Regulations for Preventing Collisions at Sea |
| DAC | Digital to analog converter |
| DCU | Dynamic control unit |
| DHT | Deep Hough Transform |
| GAN | Generative Adversarial Network |
| GNSS | Global Navigation Satellite System |
| GPS | Global Positioning System |
| IALA | The International Association of Marine Aids to Navigation and Lighthouse Authorities |
| IHO | International Hydrographic Organization |
| IoU | intersection over union |
| IMO | International Maritime Organization |
| IMU | Inertial Measurement Unit |
| IPsec | IP Security Architecture |
| kVA | kilovolt-amperes |
| LiDAR | Light detection and ranging |
| LiFePO4 | Lithium Iron Phosphate |
| mAP | mean Average Precision |
| MASS | Maritime Autonomous Surface Ships |
| MMSI | Maritime Mobile Service Identity |
| MQTT | Message Queuing Telemetry Transport |
| NGC | Navigation, guidance, and control |
| PCAP | Packet Capture |

| PoE | Power over Ethernet |
|---|---|
| RGB | Red, Green, and Blue |
| RHT | Reverse Hough Transform |
| ROC | Remote Operations Center |
| ROS2 | Robotic Operating System 2 |
| RTSP | Real-Time Streaming Protocol |
| SLAM | Simultaneous Localization and Mapping |
| SOLAS | The International Convention for the Safety of Life at Sea |
| SQL | Structured query language |
| Traficom | Finnish Transport and Communications Agency |
| Turku UAS | Turku University of Applied Science |
| TDSS-G1 | The Turku UAS DeepSeaSalama—GAN dataset 1 |
| UAV | Unmanned aerial vehicle |
| UNCLOS | The United Nations Convention on the Law of the Sea |
| USV | Unmanned Surface Vessel |
| VAC | Volts Alternating Current |
| VDC | Volts Direct Current |
| VDES | VHF Data Exchange System |
| ViT | Vision Transformer |

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
