# Peer review of "Deep Learning Test Platform for Maritime Applications: Development of the eM/S Salama Unmanned Surface Vessel and Its Remote Operations Center for Sensor Data Collection and Algorithm Development"

_remotesensing, doi:10.3390/rs16091545_

Round 1

Reviewer 1 Report

Comments and Suggestions for Authors

Summary:
The proposed article emphasises the importance of deep learning algorithms for autonomous operating maritime vessels and concludes that more suitable training data, especially multi modal data, is required to develop such algorithms and thus a platform for collecting this data is needed.

This is followed by a survey on a test platform for autonomous maritime operations consisting of an unmanned surface vehicle (UAV) and a control centre for remote operation. The system is described in detail covering topics ranging from properties of the boat used as UAV over on-board electronics and sensors to IT- and software infrastructure. This covers not only technical aspects but also legal issues. The status of the implementation of this platform is depicted and an outlook is given, which enhancements are planned for the near and not so distant future.

The methods for data collection, annotation and enhancement using GANs are outlined and a locally developed algorithm for object detection and tracking is applied to one of the proposed data sets.

The discussion focusses on the potential of the project for development of autonomous maritime vessels and deep learning algorithms in that field and gives an outlook on possible future activities in that field.

Comments:
As the article emphasises on a detailed description of the system, covering requirements and specifications, the listed components shall not only be named, but a reference to the manufacturer shall be given to complete the information on the components. E.g. line 327: which Software in detail? If it is important that specifically a “Hatteleand touchscreen” (line327) is used, a reference to product details of the manufacturer shall be given also. This applies to many components mentioned throughout the article.

The chosen detail-level is strength and weakness at the same time. For readers outside Finland the details of the certification process are less relevant and do not contribute to the relevant information on the contribution this test environment offers to researchers. Merely the last paragraph (lines 406 through 409) seems to be relevant, as it narrows the scenarios in that sensor data can be acquired (and provided). Please consider shortening the section 3.1.4.

Abbreviations and Acronyms shall be introduced and when referring to some standardized file format, protocol or the like, please add a reference to its specification. Examples are: Line 582 PCAP file, line 594 PostgreSQL, line 853 S-100 standard. If already listed in the abbreviations list, add the reference to its specification there.

The discussion section misses the mentioning of the deep learning and GAN algorithms discussed in the section before. This shall be added and the repetition of motivation and the repeated emphasis of the importance of deep learning in the maritime sector shall be reduced.

The standpoint of the authors that deep learning is crucial for marine automation and training data is needed etc. is not only given in the introductory part to show the motivation for the project, but repeated (complete or in parts) more than once throughout the article. Please remove such repetitions.

Weakness of the proposal is the attempt to cover many areas of research and engineering, ranging from UAV design and registration over Software design for data acquisition to methods of deep learning research and algorithm development, in a single article. All the covered aspects related to the UAV are well connected to one article and result in a contribution that addresses a wide group of readers. However, any single reader will be most likely only interested in a small subset of the topics covered. Splitting this article into smaller, separate ones would address the respective readers much better.

Please consider enlarging the text in the figures, it is hardly readable at a normal zoom factor.

Author Response

Dear Reviewer,

we appreciate your time and effort in reviewing our manuscript. Your comments and suggestions are valuable in enhancing the quality of our work. We have carefully considered each point and have made revisions accordingly. Here is a summary of the changes:

  1. Component Details:

In response to your feedback, we have revised our article to include references for the components where the model or manufacturer has a significant impact on the system’s performance. For instance, we have added references for the GPS compass, as the performance of this device is critical to the system’s functionality.

However, we believe that including references for every single component may not always add value to the article, especially when the specific model or manufacturer does not significantly affect the system’s performance or the understanding of the system. For example, the VHF radio used in our system is a standard component, and its specific model or manufacturer does not influence the system’s performance. Therefore, we have decided not to include references for such components and have removed the specific models from the text. For other components, the relevant references have been added. For the examples mentioned by the referee on line 327, the specific product and brand names were removed.

  1. Certification Process: We understand that the details of the certification process may not be relevant to all readers. Therefore, we have shortened section 3.1.4, fully retaining only the last paragraph (lines 406 through 409) which is most relevant to the scenarios in which sensor data can be acquired. Everything before that is condensed to a smaller length. This condensed text is highlighted in the document.

  1. Abbreviations and References: We have introduced the full name for PCAP and added references to the specifications and standards for it and PostgreSQL and S-100. PostgreSQL is described in more detail and the S-100 standard is mentioned with the full name and a reference.
  1. Deep Learning and GAN in the discussion Section: We have expanded the discussion section by adding a paragraph focusing on the deep learning and GAN algorithms discussed in the preceding section. Additionally, we have included four new references that directly relate to the content of this paragraph.

  1. Repetitions: We have removed several repetitions of the standpoint that deep learning and training data is crucial for marine automation from sections 3, 4, and the Conclusions. In the sections where we have opted to retain the mentions of deep learning and/or training data, we believe they provide context for the particular work described and are essential.

  1. Scope of the Article: We appreciate your suggestion to divide the article into smaller, focused pieces for different readers and understand perfectly your viewpoint. While we carefully considered this approach already during the article drafting, we opted to present an overview of our full test platform at a relatively high level. This allows readers to understand the overall concept of our test platform and its potential applications. Additionally, the idea was that this article would serve as an informative reference for future publications that will delve into specific aspects of the platform (hardware or software), aimed to a smaller subset of readers. Furthermore, this article would serve as a main reference for articles related to the collected data and algorithms developed for USV autonomous operations.
  1. Text in Figures: We agree that the figures were really difficult to read in the previous draft. Here’s a concise summary of the improvements made to the figures to make them more easily readable with normal zoom factor:

Figures 1, 3, 6, 7, 9, 10, 12, 13, and 20: Many of these figures were initially small and challenging to read. However, following the reviewers’ comments and suggestions, we have updated these figures. The font size has been increased, and we’ve also adjusted the composition where necessary. As a result, their readability has significantly improved.

Figures 1, 12, 13, 15, 17, 18, and 20: Additionally, we’ve widened the image width for these figures to enhance readability. The larger dimensions should make it easier for readers to engage with the content.

We believe these revisions have greatly improved the manuscript and hope that it is now suitable for publication. Thank you again for your constructive feedback!

Best Regards,

Juha Kalliovaara

Reviewer 2 Report

Comments and Suggestions for Authors

We appreciate the efforts of the authors’ contributions regarding USV with deep learning applications.

Appreciation:

1. Development of the USV with integration of various sensors such as LiDAR and RGB camers for the analysis of the objects

2. The experiments were conducted using Singapore maritime databases.

Comments.

1. Table 4: Related information is not included in the conclusion.

2. Figure 19 is part of the USV or virtual results , explain clearly

3. Computation was used for the complete deployment system, as mentioned in Section 3.1, but its power consumption and other technical information were not mentioned.

Author Response

Dear Reviewer,

We are grateful for your time and effort in reviewing our manuscript and the appreciation given!

Your comments and suggestions are invaluable in improving the quality of our work.

Having considered each point you raised, we have made the necessary revisions. Below, we outline the specific changes implemented:

  1. Table 4: We have included discussion on the relevance of the mAP results from Table 4 in the conclusions. There was a sentence already stating this, but it was a little unclear. It has been now greatly elaborated and improved, thus covering much better the significance of the results in Table 4.
  2. Figure 19: We are duly sorry for the confusion. It is not related to the USV or virtual results. The object tracking method is applied to a video sourced from the Singapore Maritime Dataset. This dataset comprises 81 existing and freely accessible videos related to maritime scenarios. This is now clarified in the text.
  3. Computation Details: We have enhanced Section 3.1.1 by incorporating details related to the power consumption and the technical aspects of computation utilized in our existing ICT and data collection system. Additionally, we have included information about upcoming updates that will be implemented to support autonomous navigation.

We believe these revisions have improved the manuscript and hope that it is now suitable for publication. Thank you again for your constructive feedback!

Best Regards,

Juha Kalliovaara

Reviewer 3 Report

Comments and Suggestions for Authors

Generally, this work is interesting and provides informative knowledge. However, there are many aspects to be well accommodated as follows:

1. The overall presentation is rather tedious and is highly required to be tailored.

2. The title seems inconsistent with the content since deep learning techniques are actually not unfolded in this work. From my viewpoint, it mainly exhibits a test platform.

3. Figures 1, 3, 7, 9, 10 and 13 are required to be made much more informative.

4. If the ROC development is an contribution point within this work, it should be formulated in details rather than architecture description.

5. Similarly, it can hardly observe the contribution from the multi-modal data platform which should be unfolded with innovations.

6. It seems the open-access dataset is rather limited with only 3781 samples supporting a data platform. The authors may refer to a benchmark: 10.1016/j.apor.2023.103835.

7. The discussion is pretty well by connecting to promising potentials, and can be made informative by incorporating state-of-the-art works, for instance, 10.1109/TMECH.2021.3055450 (heterogeneous swarm), and 10.1109/TITS.2024.3364770, 10.1109/TII.2020.3033794 (GPS-denied navigation and control).

8. The conclusion has to be condensed.

Comments on the Quality of English Language

It's OK but needs to polish.

Author Response

Dear Reviewer,

Thank you for your thoughtful review of our scientific article submitted to the MDPI Remote Sensing journal.

We appreciate your valuable feedback and have taken each of your comments into consideration.

Below, we address the issues raised:

  1. Overall Presentation: We acknowledge that the initial presentation was somewhat tedious. We have revised the manuscript to make it more concise by removing some repetitive information mentioned more than one time in different sections (for example, the importance of multi-modal sensor data was mentioned several times) and shortening some sections like 3.1.4 Commercial Craft Certification in Finland and 6 Conclusions, ensuring a smoother reading experience for our audience.
  2. Title Consistency: You rightly pointed out that the title did not align perfectly with the content. We have adjusted the title to better reflect the focus of our work, emphasizing the test platform aspect rather than deep learning techniques. The revised title is “Deep Learning Test Platform for Maritime Applications: Development of the eM/S Salama Unmanned Surface Vessel and its Remote Operations Center for Sensor Data Collection and Algorithm Development.”
  3. Figures Enhancement: Figures 1, 3, 7, 9, 10, and 13 have been revisited as suggested. The text has been made bigger and the information content has been improved. The same was done also figures 6 and 12. All the other figures have also been rechecked and the text and blocks have been updated where necessary to make them more informative and the text easier to read. The width was increased in the document for figures 1, 12 and 13 to further improve readability.
  4. ROC Development: We appreciate your feedback on the ROC development section. Our main focus and contribution in the ROC development has been on the dynamic control unit, whose development has been described in more detail. The ROC software was procured from a commercial supplier, so it is described on detail level which gives an understanding on how it functions and how the dynamic control unit is integrated into the system. This is now stated in the article: "Our main focus and contribution in the ROC development has been on the DCU, whose development is described in more detail. The ROC itself is described in the article on a detail level which gives an understanding on how it functions and how the DCU is integrated into the overall ROC system."
  5. Multi-Modal Data Platform: We agree that to have a stronger contribution from the multi-modal data platform, it needs further elaboration. However, due to excessive length we already have in this article, we plan to do this in separate publications in the future where we could cite to this paper with base information on the test platform and concentrate on a more elaborated presentation of the multi-modal data platform and the overall multi-modal data collection system, including the synchronization methods between many different modalities and the search capabilities within the data platform.

  1. Dataset Size: We have now referred to the benchmark given (Marine vessel detection dataset and benchmark for unmanned surface vehicles). A declaration that the number of images in a dataset and their diversity in terms of for example different viewing angles, lighting conditions, occlusions, scales and hull parts is very important for the performance of object detection models is added, along with a note that the dataset we published here is relatively small and mainly used to showcase how the performance of object detection can be improved by generating additional synthetic images to train the models.

  1. Discussion and State-of-the-Art: We have enriched the discussion section by incorporating the provided references to state-of-the-art works. Specifically, we have cited the following articles:

He, H., Wang, N., Huang, D., & Han, B. (2021). “Active Vision-Based Finite-Time Trajectory-Tracking Control of an Unmanned Surface Vehicle Without Direct Position Measurements.” IEEE Transactions on Intelligent Transportation Systems.

Wang, N., & Ahn, C. K. (2021). “Coordinated trajectory-tracking control of a marine aerial-surface heterogeneous system.” IEEE/ASME Transactions on Mechatronics, 26(6), 3198-3210.

Wang, N., & He, H. (2021). “Extreme Learning-Based Monocular Visual Servo of an Unmanned Surface Vessel.” IEEE Transactions on Industrial Informatics, 17(8), 5152-5163.

  1. Concluding Remarks: The conclusions have been condensed to about a half of the original, ensuring a succinct summary of our findings. Some repetitive parts to previous  chapters have been completely removed. The modified text is highlighted.
  2.  

Once again, we appreciate your thorough review, and we believe these revisions significantly enhance the quality of our manuscript. We hope you find the updated version more engaging and informative!

Best Regards,

Juha Kalliovaara

Reviewer 4 Report

Comments and Suggestions for Authors

An excellent clearly written paper. The overall review of the state of subject area is useful. The depth of detail of your technical work is extremely useful. Very helpful to have the detail reported openly as many other developments are considered to be commercially sensitive.  The inclusion of the discussion on regulatory issues is a valuable addition to the technical descriptions. 

It would be great if you could produce updates on your progress. These need not be as detailed or broad is this paper -- which will have been a lot of work.

Author Response

Dear Reviewer,

We are grateful for your time and effort in reviewing our manuscript and we appreciate your positive feedback.

We are planning to publish further updates on our research journey towards making our test vessel autonomous in research articles and we also plan to create a web page, which would act as a central hub for finding our published datasets.

Do not hesitate to contact us if you want some direct updates on the status of our work!

Best Regards,

Juha Kalliovaara

Reviewer 5 Report

Comments and Suggestions for Authors

Summary

·         This paper presents details of a test platform for advancing autonomous maritime operations, featuring the unmanned surface vessel eM/S Salama and a remote operations center. The importance of collecting and annotating multi-modal sensor data from the vessel is emphasized, crucial for developing deep learning algorithms enhancing situational awareness and guiding autonomous navigation.

Overall Review

·         Literature Review: Please consider reviewing the following related papers and add to the paper:

o   Dong, Yuxin, et al. "Ship object detection of remote sensing image based on visual attention." Remote Sensing 13.16 (2021): 3192.

o   Su, Li, et al. "A survey of maritime vision datasets." Multimedia Tools and Applications 82.19 (2023): 28873-28893.

o   Lisbon, P. T. "Ship Segmentation in Areal Images for Maritime Surveillance."

o   Sun, Bowen, et al. "Automatic ship object detection model based on YOLOv4 with transformer mechanism in remote sensing images." Applied Sciences 13.4 (2023): 2488.

o   Vidimlic, Najda, et al. "Image Synthesisation and Data Augmentation for Safe Object Detection in Aircraft Auto-landing System." VISIGRAPP (5: VISAPP). 2021.

·         Reproducibility of Results: The reproducibility of results is under question as code is not available.

·         Figures/Tables:

o   Resolution of figures should be increased.

o   Please increase the size of texts in the figures.

o   Figure 16: use the same size for all figures.

o   Figure 13: text overlaps with the boarder of a box.

·         Results:

o   Table 4: Please report the results of mAP for large, medium, and small targets.

o   Figure 16: synthetic night images do not look realistic compared to the generated day and sunset images. Does the utilized method suffer from nighttime synthetic maritime image generation?

Comments on the Quality of English Language

Writing: Overall, the paper is well-written and easy-to-follow. There are some typos in the text. I would recommend authors revising the entire paper and fixing all typos.

§  Figure 14(a): “Captured image from Thermal camera.” -> ”Captured image from thermal camera.”

Author Response

Dear Reviewer,

Thank you for your thoughtful review of our article submitted to the MDPI Remote Sensing journal.

We appreciate your positive feedback and constructive comments.

We have carefully addressed each of the issues you raised, and we are pleased to inform you of the following updates:

Literature Review: We have incorporated all the suggested related papers into our article. Specifically, we have included mentions of ship object detection based on satellite imagery, introduced the survey on maritime vision datasets, introduced ship detection based on aerial images, mentioned synthetic data generation in more manual fashion along with the related reference, and added the reference for automatic ship object detection models based on transformer mechanisms. These additions further enhance the context and depth of our work.

Reproducibility of Results: We apologize for the omission regarding code availability. Regarding the deep learning algorithms, we have presented just a brief introduction to each algorithm in this article to provide context on the deep learning algorithms that can be used with the test platform data. Regarding the overall test platform system, the code often relies on specific system configurations, making it hard to share without the full setup. Also, the code is constantly evolving in iterations according to the upcoming needs for improvements, so frequent changes would make the shared code outdated. For the future, we will however think of feasible methods for code sharing through for example Github. Especially the autonomous navigation algorithms which are more independent from the specific sensors and devices will be shared along the future articles about them.

Figures/Tables:

    • We have increased the resolution or size of figures, ensuring better clarity and readability.
    • Text sizes within the figures have been adjusted to improve readability.
    • Figure 16 now maintains consistent sizing across all components.
    • The issue with text overlapping in Figure 13 has been resolved.

Results:

    • In a new table, Table 5, we have provided the mAP (mean Average Precision) values for large, medium, and small targets, as requested. A descriptive text of these results is added to the article.
    • Regarding Figure 16, we acknowledge the discrepancy in the synthetic night images. Our method indeed faces challenges in generating realistic nighttime maritime images. We are using a pretrained GAN model to generate synthetic data, and it is a good suggestion to add more maritime and nighttime samples to enhance its ability to generate more realistic data for nighttime mode. The existing results underscore the importance of future work in this area, specifically focusing on further development of GAN models within the maritime domain.

Quality of English Language:

    • We appreciate your keen eye for detail. The entire paper has undergone thorough proofreading, and all typos we found have been corrected.
    • Figure 14(a) now accurately reads: “Captured image from thermal camera.”

Once again, thank you for your valuable input. We remain committed to enhancing the quality and impact of our research.

Best Regards,

Juha Kalliovaara